# DIFFUSION FINE-TUNING:
# ITERATIVE REFINEMENT FOR ADVANCED GROUNDING WITH DIFFUSION LARGE LANGUAGE MODELS

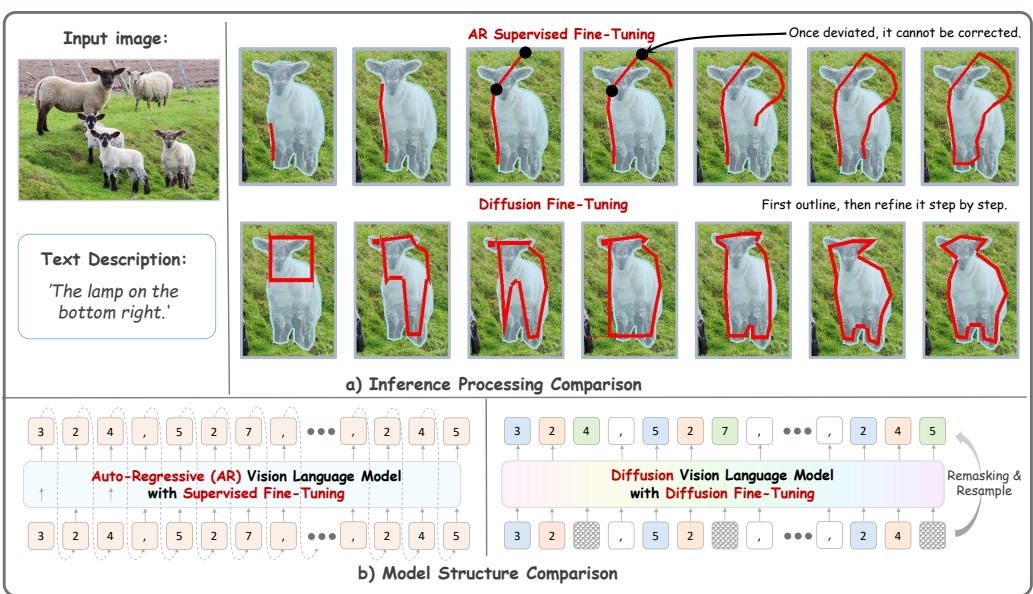

Figure 1: **Autoregressive (AR) models vs. Diffusion Fine-Tuning (DFT).** While an AR model generates vertices sequentially, making early errors irreversible, DFT iteratively refines a coarse outline to prevent such error accumulation. AR uses unidirectional sequence generation, while the DFT optimizes all vertices in parallel.

## ABSTRACT

While Large Vision-Language Models (LVLMs) excel at simple bounding box grounding, they reveal fundamental limitations in tasks requiring precise spatial localization, most notably polygon grounding. We identify this bottleneck as stemming from two fundamental flaws of the autoregressive (AR) paradigm: 1) irreversible error accumulation, where early vertex errors propagate uncorrected through the sequence; and 2) a lack of global planning, which leads to a suboptimal allocation of the finite number of vertices (16 points) on the object's contour. We propose Diffusion Finetuning (DFT) to reframe visual grounding as a robust, parallel global optimization. Its core is a 'sculpture-like', coarse-to-fine generation process, where coordinate digits are predicted hierarchically (e.g., hundreds, then tens, then units) to progressively refine the shape from a coarse outline to precise details. We use a novel Hierarchical Curriculum Learning strategy that progressively refines the loss supervision, guiding the model from a rough outline to a precise delineation. Extensive experiments show that DFT achieves state-of-the-art on both 2D bounding box and 16-point polygon grounding, and demonstrates its strong performance on the complex 9-DoF 3D bbox grounding task.

## 1 INTRODUCTION

While Large Vision-Language Models (LVLMs) such as GPT-4o (OpenAI, 2024) and LLaVA (Liu et al., 2023) have achieved remarkable success in simple bounding box grounding tasks, their performance reveals significant shortcomings on tasks requiring higher spatial precision, such as Polygon

Figure 2: **DLM Training.** Direct DLM training (left) fails to converge, producing a coarse, invalid output with overlapping contours. Right part displays the outputs from the three progressive stages of Hierarchical Curriculum Learning (HCL): from a coarse outline, to a refined contour, and finally to the detailed shape.

Grounding. This task requires the model to use global planning to accurately outline an object with a limited vertex sequence, severely testing its spatial understanding.

The root of this bottleneck is the **auto-regressive (AR)** paradigm (Bai et al., 2025; Chen et al., 2024) used by most leading LVLMs. These decoder-only models generate vertices sequentially, with each new vertex prediction conditioned solely on the ones before it. This sequential process has two fundamental flaws. First is irreversible error accumulation: an early prediction error (the fourth point in the sheep example in Figure 1) triggers a cascade of mistakes that permanently and severely distort the final shape. Second, a failure in global planning: the 'short-sighted' model cannot perceive the full contour to rationally allocate its vertices, often overspending vertices on smooth areas and lacking enough for complex details like the sheep's horns, resulting in a coarse outline. Both flaws limit its localization accuracy.

To address these issues, we introduce the **Diffusion Language Model (DLM)** (Nie et al., 2025; Yang et al., 2025). Unlike AR's sequential method, its iterative optimization process starts from a masked state to globally correct all vertices in parallel, directly resolving the error accumulation problem. However, the parallel generation of a complete polygon (16 vertices) is non-trivial for conventional training methods, which fail to converge and produce only a rough outline of the sheep as shown on the left of Figure 2. This indicates that a specialized training and fine-tuning strategy is required to unleash the potential of diffusion models for high-precision localization fully.

To overcome the aforementioned challenges, we propose the **Diffusion Fine-tuning (DFT)** framework. The core idea is to guide the diffusion model to exhibit a 'sculpture-like', coarse-to-fine optimization process during its iterative denoising inference. As shown in Figure 2 (right), this coarse-to-fine mechanism is exemplified by three-digit coordinate quantization, where we guide the model to progressively determine the hundreds (for the rough outline), tens (to refine the shape), and units digits (for precise delineation). This hierarchical strategy simplifies the complex prediction problem by decomposing it into sub-tasks, which improves training stability and final localization accuracy. To achieve this coarse-to-fine inference, we design a three-stage **Hierarchical Curriculum Learning (HCL)** strategy. This strategy progressively increases the loss granularity, supervising the hundreds digit (for the macro-contour), then the tens digit (to optimize structure), and finally all three digits (for precise delineation).

Extensive experiments validate its effectiveness. To evaluate the challenging task of polygon grounding, we first introduce **RefCOCO-Polygon**, a new 16-point benchmark from the RefCOCO (Mao et al., 2016) dataset. On this demanding benchmark, our method achieves state-of-the-art (SOTA) performance. Meanwhile, on standard bounding box benchmarks like RefCOCO, DFT sets a new state-of-the-art by significantly outperforming top autoregressive models. To further verify the framework's versatility, we extended it to more complex 3D scenarios. We built the first referring 3D grounding benchmark, **Ref3D**, by integrating leading indoor (Objectron (Adel Ahmadyan, 2021), SUN-RGBD (Song et al., 2015), ARKitscene (Baruch et al., 2021), and Hypersim (Roberts et al., 2021)) and outdoor (KITTI (Geiger et al., 2013) and nuScenes (Caesar et al., 2020) datasets. Experimental results show that our method maintains its strong performance on the challenging 9-Degree-of-Freedom (9-DoF) 3D bounding box grounding task.

The main contributions of our work are as follows:

- **Diffusion Fine-Tuning**: We propose a new paradigm that reframes visual grounding as a parallel global optimization problem, using a 'sculpture-like', coarse-to-fine process.

- **Sota Performance for Visual Grounding**: Experiments demonstrate that DFT achieves SOTA performance on both 2D bounding box and difficult 2D polygon grounding tasks.

- **Versatility in Complex & 3D Tasks**: We further demonstrate its versatility by extending it to the complex 9-DoF 3D bounding box grounding task and achieving excellent results.

## 2 RELATED WORK

**LVLMs For Grounding.** The visual grounding capabilities of Large Vision-Language Models (LVLMs) have evolved significantly from holistic semantic understanding to fine-grained region-level localization. Initial efforts, such as BuboGPT (Zhao et al., 2023), adopted an 'off-the-shelf module' strategy, relying on separate visual models to locate entities, which limited deep cross-modal fusion. To achieve tighter vision-language integration, mainstream research has rapidly shifted towards end-to-end solutions that make grounding an inherent model capability. These solutions primarily follow two technical paradigms: the 'pixel-to-sequence' (pix2seq) approach, seen in models like Kosmos-2 (Peng et al., 2023) and GroundingGPT (Li et al., 2024), which serialize coordinates into text tokens for unified prediction; and the 'pixel-to-embedding' (pix2emb) approach, pioneered by NExT-Chat (Zhang et al., 2023), which outputs location embeddings that can be decoded into bounding boxes or even pixel-level masks, as demonstrated by GLaMM (Rasheed et al., 2024). Furthermore, addressing the demands of dynamic scenes like games, cutting-edge works such as VTimeLLM (Huang et al., 2024) have successfully extended grounding from the spatial to the temporal domain across video and audio, laying a solid foundation for precise interaction in complex multimodal environments.

**Multi-Modal Diffusion Language Models.** To overcome the slow, unidirectional limitations of autoregressive (AR) models (OpenAI, 2024; Bai et al., 2025), researchers have applied diffusion language models (DLMs) (Li et al., 2022) for parallel and bidirectional text generation using techniques like masked diffusion (Austin et al., 2021; Nie et al., 2025; Gong et al., 2025; Jiang et al., 2025; Li et al., 2025). Building on their success in text, diffusion models were rapidly extended to multimodal tasks by using cross-modal conditions. In the vision-language domain, LLaDA-V (You et al., 2025) pioneered this approach for VQA tasks, which paved the way for more general multimodal large diffusion language model frameworks (Cui et al., 2025; Yang et al., 2025; Song et al., 2025). This methodology has also been quickly adopted in specialized fields such as biomedical image understanding (Dong et al., 2025) and embodied intelligence (Liang et al., 2025). We propose the Diffusion Fine-tuning (DFT) framework to adapt diffusion models for high-precision spatial localization, an area less explored than semantic tasks. Our 'coarse-to-fine' strategy achieves excellent results on the 16-point polygon grounding task.

## 3 METHODOLOGY

In this section, Subsection 3.1 formulates the visual grounding task and its challenges. Subsection 3.2 introduces our core Hierarchical Coordinate Decomposition (HCD) representation. Subsection 3.3 then presents the Diffusion Fine-Tuning model built upon HCD, and Subsection 3.4 describes the Hierarchical Curriculum Learning strategy used to train it effectively.

### 3.1 SEQUENTIAL REPRESENTATION OF POLYGON GENERATION

The core task of visual grounding is to accurately locate a target object in an image $I$ based on a given textual instruction $T_{ins}$. Traditional methods often rely on predicting bounding boxes, which struggle to delineate irregularly shaped objects. To achieve more flexible and precise localization, we adopt polygons to outline the target's contour. To adapt this to a sequence generation framework, we transform the task of localizing the target polygon into a conditional sequence generation problem. Specifically, a polygon with $N$ vertices can be uniquely defined by its corresponding one-dimensional coordinate sequence $S_0$:

$$S_0 = (x_1, y_1, x_2, y_2, \ldots, x_N, y_N), \tag{1}$$

Here, $(x_i, y_i)$ represents the coordinates of the $i$-th vertex of the polygon. We normalize the image dimensions and quantize each coordinate value into a three-digit integer, such that $x_i, y_i \in \mathbb{Z}_C = \{000, 001, \ldots, 999\}$. Consequently, the objective of the entire visual grounding task is to train a model $p_\theta$, parameterized by $\theta$, that can generate the most probable target coordinate sequence $S_0$ given the image $I$ and the textual instruction $T_{ins}$. This is mathematically expressed as maximizing the following log-likelihood: $\max_\theta \log p_\theta(S_0 \mid I, T_{ins})$. However, directly modeling the coordinate sequence $S_0$ as described above presents two significant challenges:

- **Large Vocabulary Space and Difficulty in Learning Numerical Relationships:** Treating each three-digit coordinate (from '000' to '999') as a distinct token results in a massive vocabulary of 1000 tokens. This not only dramatically increases the model's output complexity but, more critically, makes it difficult for the model to learn the inherent numerical relationships between these discrete tokens. For instance, the model would not know a priori that '501' and '502' are numerically close, while being distant from '100'.

- **Lack of Essential Structural Priors:** This naive serialization method treats all coordinate tokens as equidistant and independent, completely ignoring the geometric continuity and spatial locality inherent in the coordinates themselves. The model perceives the coordinates merely as undifferentiated symbols rather than as points in a two-dimensional space. This contradicts the physical nature of the visual grounding task and hinders the model's ability to effectively learn spatial structures.

To overcome these challenges, a new coordinate representation is needed—one that can both reduce the vocabulary size and integrate crucial structural priors into the model's design. In the next section, we will introduce our proposed **Hierarchical Coordinate Decomposition** strategy to address this.

### 3.2 HIERARCHICAL COORDINATE DECOMPOSITION (HCD)

We propose the **Hierarchical Coordinate Decomposition (HCD)** strategy to address the aforementioned representational bottleneck. Its core idea is to decompose each scalar coordinate value by its decimal places, transforming a single coarse-grained prediction task into multiple fine-grained ones with an intrinsic hierarchical relationship. We define a decomposition function $\Psi : \mathbb{Z}_C \rightarrow \mathbb{Z}_{10}^3$, which maps any coordinate $c \in \mathbb{Z}_C$ to its hundreds, tens, and units digits (calculated as $\lfloor c/100 \rfloor$, $\lfloor (c \pmod{100})/10 \rfloor$, and $c \pmod{10}$ respectively). Applying $\Psi$ to the full coordinate sequence $S_0$ reconstructs it into three parallel digit sequences (each of length $2N$): the high-bit sequence $S_0^H$ (hundreds), mid-bit sequence $S_0^T$ (tens), and low-bit sequence $S_0^U$ (units).

This decomposition offers two key advantages. It drastically reduces the prediction vocabulary from $\mathbb{Z}_C$ (1000 tokens) to the compact $\mathbb{Z}_{10}$ (10 digits), simplifying learning. It also embeds a 'coarse-to-fine' structural prior, where the sequences $S_0^H$, $S_0^T$, and $S_0^U$ hierarchically refine the object's contour from a general outline to its precise details. For example, a coordinate like $(258, 703)$ is hierarchically decomposed into a coarse sequence $S_0^H = (2, 7)$, a refining sequence $S_0^T = (5, 0)$, and a precise sequence $S_0^U = (8, 3)$. This approach reframes the task into jointly modeling three simpler, hierarchically dependent digit sequences. Next, we introduce the hierarchical diffusion model tailored for this representation.

### 3.3 DIFFUSION FINE-TUNING

#### 3.3.1 PRELIMINARY: DISCRETE DIFFUSION IN CONTINUOUS TIME

Our model is based on a continuous-time ($t \in [0, 1]$) discrete diffusion process using an absorbing state. The forward process, $q$, gradually transitions an initial one-hot encoded digit token, $x_0$, into the absorbing `[MASK]` token, $m$. We employ a linear noise schedule, where a token remains as $x_0$ with probability $\alpha_t = 1 - t$ and transitions to $m$ with probability $1 - \alpha_t = t$. The transition probability to a noisy state $x_t$ is therefore a Categorical Distribution whose probability vector is a linear interpolation of $x_0$ and $m$:

$$q(x_t|x_0) = \text{Cat}(x_t; \alpha_t x_0 + (1 - \alpha_t)m), \tag{2}$$

This formula intuitively shows $x_t$ is original $x_0$ with probability $\alpha_t$ and `[MASK]` token $m$ with probability $1 - \alpha_t$. Correspondingly, the reverse process trains a network $p_\theta$ to invert this noising—predicting clean $x_0$ from noisy $x_t$ (containing `[MASK]` tokens). For the derivation and proof of Equation (2), see Appendix A.

#### 3.3.2 FORWARD PROCESS: INDEPENDENT NOISING

The forward process of Diffusion Fine-Tuning is designed for strategic simplicity. Instead of embedding the HCD hierarchy into the noising stage, we intentionally apply noise symmetrically and independently to the three hierarchical digit sequences $S_0^H$, $S_0^T$, and $S_0^U$.

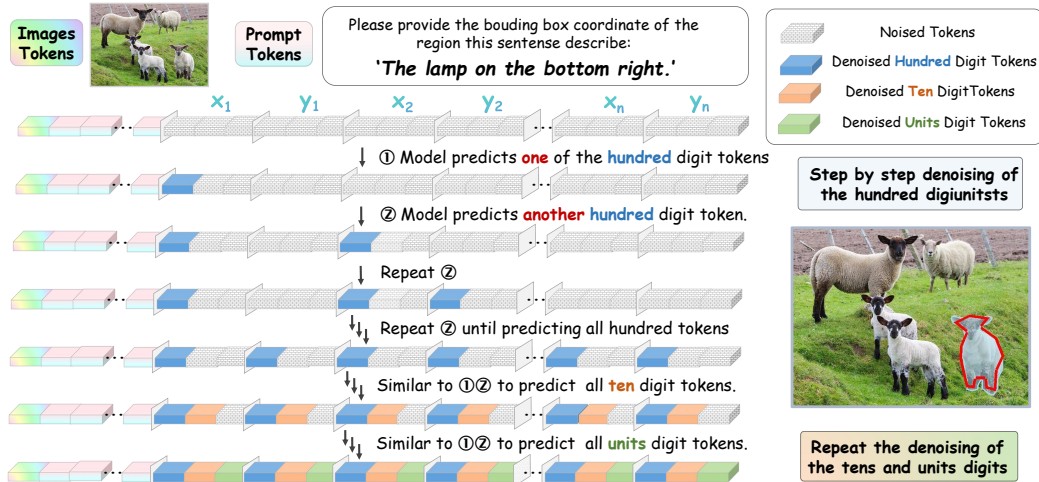

Figure 3: **The inference denoising process.** Starting from a fully masked state, it hierarchically predicts all 'hundreds', then 'tens', and finally 'units' digit tokens.

This means that for any time $t \in [0,1]$, each digit, regardless of its hierarchical level, has the same independent probability $t$ of being replaced by a $\boxed{\texttt{[MASK]}}$ token. This symmetry is crucial because it avoids giving the model any structural 'hints' or shortcuts during noising. The joint transition probability from the original sequences $S_0 = \{S_0^H, S_0^T, S_0^U\}$ to a noisy state $S_t = \{S_t^H, S_t^T, S_t^U\}$ is therefore a simple product of the individual probabilities:

$$q(S_t|S_0) = \prod_{k \in \{H,T,U\}} q(S_t^k|S_0^k), \tag{3}$$

This indicates that the noising process is performed independently, regardless of the bit. This independent noising approach not only ensures simplicity but also provides the model with diverse corrupted samples that span different hierarchical levels and noise levels during training.

### 3.3.3 REVERSE PROCESS: HIERARCHICAL & CONDITIONAL DENOISING

The inference is shown in Figure 3. The goal of reverse denoising is to use a Transformer-based model $f_\theta$ to approximate the posterior distribution $q(S_s|S_t, S_0)$. The model takes the noisy joint state $S_t$, time step $t$, and image-text conditions $(I, T_{instr})$ as inputs, and outputs the probability estimates of the original sequence at each hierarchy, denoted as $\hat{P}_0 = \{\hat{p}_0^H, \hat{p}_0^T, \hat{p}_0^U\}$. To leverage the hierarchical structure, the joint probability is decomposed into a product of 'coarse-to-fine' conditional probabilities and **the process is not independent across hierarchies**:

$$p_\theta(S_0|S_t, t) = p_\theta(S_0^H|S_t, t) \cdot p_\theta(S_0^T|S_t, t, S_0^H) \cdot p_\theta(S_0^U|S_t, t, S_0^H, S_0^T), \tag{4}$$

Specifically, the model first predicts $S_0^H$ to determine the macro outline, then predicts $S_0^T$ for refinement based on $S_0^H$, and finally predicts $S_0^U$ (combining the first two hierarchies) to achieve precise localization. For the detailed algorithm pseudocode, see Appendix B. The training strategy required to achieve this asymmetric denoising will be detailed in the next section.

### 3.4 TRAINING STRATEGY: HIERARCHICAL CURRICULUM LEARNING

Our **Hierarchical Curriculum Learning strategy** shown in Figure 4 trains the complex model dependencies $p_\theta(S_0|S_t, t)$, which are required for the inference process in the previous section. It employs an easy-to-hard approach by defining a separate cross-entropy loss, $\mathcal{L}_k$, for each hierarchy ($k \in \{H, T, U\}$), calculated only at masked positions. To enforce the hierarchical structure, we use Teacher Forcing, feeding higher-level ground-truth labels as conditions when calculating the loss for lower levels. The loss function $\mathcal{L}_k(\theta)$ for any given hierarchy $k$ is defined as:

Figure 4: **Training process and model architecture.** a) The Hierarchical Curriculum Learning (HCL) process refines the output by progressively supervising the hundreds, tens, and finally all digits. b) The model, comprising a visual encoder, text tokenizer, and DLM, which processes image and text inputs to generate coordinates.

$$\mathcal{L}_k(\theta) = \mathbb{E}_{t,S_0,S_t} \left[ -\sum_{i=1}^{2N} \mathbf{1}_{(S_t^k)_i=[\text{M}]} \log p_\theta((S_0^k)_i | \mathcal{C}_i^k) \right] \tag{5}$$

where $\mathbf{1}_{(S_t^k)_i=[\text{M}]}$ is an indicator function ensuring the loss is only computed at masked positions. The condition set $\mathcal{C}_i^k$ varies depending on the hierarchy, specifically:

$$\mathcal{C}_i^k = \begin{cases} (S_t, t, I, T_{\text{instr}}) & \text{if } k = H \\ (S_t, t, S_0^H, I, T_{\text{instr}}) & \text{if } k = T \\ (S_t, t, S_0^H, S_0^T, I, T_{\text{instr}}) & \text{if } k = U \end{cases} \tag{6}$$

Based on the above definitions, we divide the entire training process into three progressive stages, dynamically combining the total loss $\mathcal{L}_{\text{train}}$ using a piecewise function of the global training step $g$:

$$\mathcal{L}_{\text{train}}(g,\theta) = \begin{cases} \mathcal{L}_H(\theta) & \text{if } 0 \leq g < G_1 \\ \mathcal{L}_H(\theta) + \mathcal{L}_T(\theta) & \text{if } G_1 \leq g < G_2 \\ \mathcal{L}_H(\theta) + \mathcal{L}_T(\theta) + \mathcal{L}_U(\theta) & \text{if } g \geq G_2 \end{cases}$$

These three stages correspond to the different objectives of our curriculum learning:

- **Stage 1: Macro-Contour Learning** ($g < G_1$)
  In the initial training phase, only the high-bit loss $\mathcal{L}_H$ is optimized. This forces the model to disregard fine details and focus on learning the object's general position and basic.

- **Stage 2: Mid-level Structure Correction** ($G_1 \leq g < G_2$)
  After that, the mid-bit loss $\mathcal{L}_T$ is added to the total loss. With teacher forcing from $S_0^H$, the model learns to perform the first level of detailed refinement on the known coarse contour.

- **Stage 3: Fine-grained Detail Refinement** ($g \geq G_2$)
  In the final phase, the low-bit loss $\mathcal{L}_U$ is introduced. Guided by ground-truth labels of the first two hierarchies, the model learns to complete pixel-level precise localization.

## 4 EXPERIMENTS

We introduce the experimental part. Subsection 4.1 presents the experimental setups. Subsection 4.2 contains the main experimental results. Subsection 4.3 conducts a series of ablation studies.

### 4.1 EXPERIMENTAL SETUP

**Implementation Details.** Our model is initialized from the pre-trained weights of LLaDA-V (You et al., 2025). Its core architecture includes a LLaDA-8B (Nie et al., 2025) language tower, a SigLIP2 (Tschannen et al., 2025) vision encoder, and a two-layer MLP projector that maps visual features into the language embedding space. We fine-tune the model for the advanced visual grounding task, employing the proposed Hierarchical Curriculum Learning strategy. It should be noted that we use dynamic resolution, which is automatically adjusted to a patch size of 384. Specifically, we use the AdamW optimizer with an initial learning rate of 1e-5 for all parameters and a cosine decay schedule, conducting the entire process on 4 NVIDIA A800 80GB GPUs. Following the curriculum

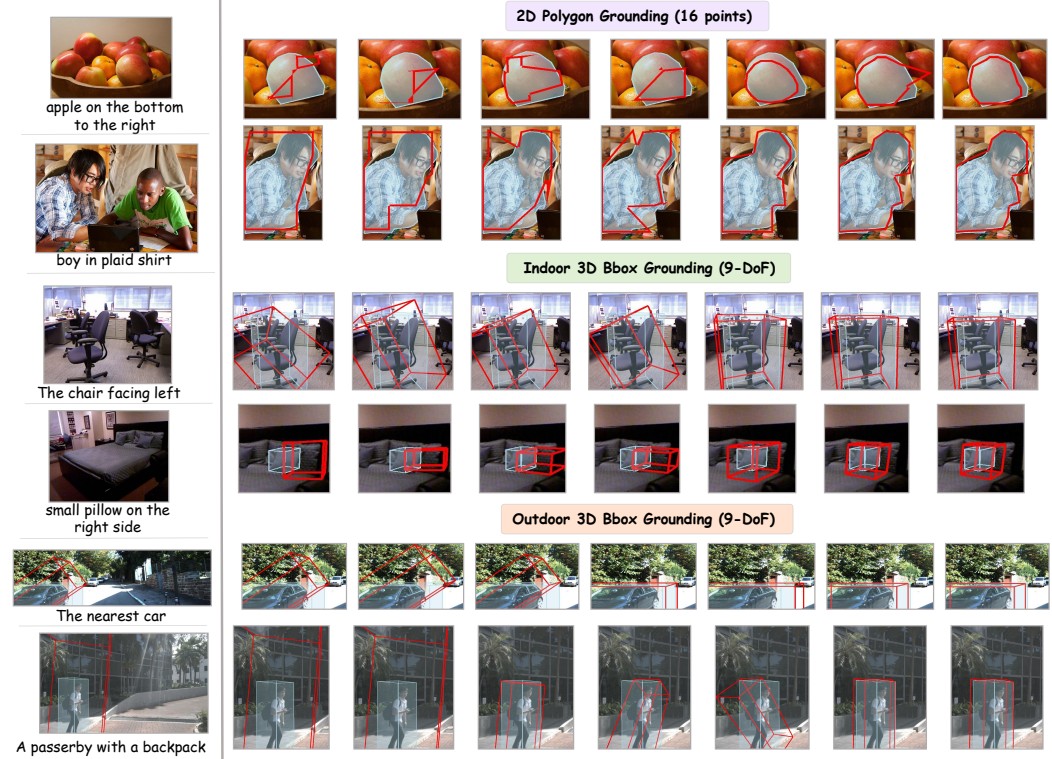

Figure 5: **Quantitative Results.** The model iteratively optimizes several tasks: 2D polygon localization (16 vertices) and 3D bounding box localization for both indoor and outdoor scenes (9-DoF).

plan, the first stage, second stage, and third stage are each trained for 1 epoch, with each stage taking approximately 7 hours.

**Datasets.** To comprehensively evaluate our proposed DFT framework, we conduct extensive experiments on several visual grounding benchmarks. For bounding box grounding, we use three widely-adopted standard datasets: RefCOCO, RefCOCO+, and RefCOCOg (Mao et al., 2016). For the more challenging task of polygon grounding— which demands finer-grained shape-aware localization compared to bounding box regression—we evaluate exclusively on a self-constructed dataset, **RefCOCO-Polygon** . Derived from the classic RefCOCO benchmark, it supplements each target object instance with dense polygon labels—including both 8-point and 16-point representations—to support the analysis of model performance across different shape granularity levels. For monocular 3D object grounding—an extension to ensure our DFT framework's generalization: indoor datasets include ARKitScenes (Baruch et al., 2021), SUNRGBD (Song et al., 2015), and hypersim (Roberts et al., 2021); outdoor datasets cover KITTI (Geiger et al., 2013), nuScenes (Caesar et al., 2020), and objectron (Adel Ahmadyan, 2021). Based on these, we construct the custom **Ref3D** dataset to focus on single-object monocular 3D grounding: its training set has 150k single-object images, test set has 1k single-object images, with specific details in the Appendix F.

## 4.2 MAIN RESULTS

We evaluated our DFT framework on three grounding tasks: 2D Bounding Box (Acc@0.5), 2D Polygon (Acc@0.1), and 3D Object Grounding (Acc@0.3). The architecture of our DFT model is adapted from a LLaVA OneVision (Zhang et al., 2024) like autoregressive model. We therefore use it as a baseline to measure the performance gains of our diffusion paradigm.

**Bounding Box Grounding.** We first evaluated our model's bounding box grounding performance on three commonly used benchmark datasets: RefCOCO, RefCOCO+, and RefCOCOg. As shown in Table 2, our DFT model performs better than or comparably to current state-of-the-art autoregressive models across all test sets. This confirms that DFT's fundamental grounding is robust, providing a solid foundation for more complex localization tasks.

| Polygon Grounding | Size | RefCOCO | | | RefCOCO+ | | | RefCOCOg | |
|---|---|---|---|---|---|---|---|---|---|
| | | val | testA | testB | val | testA | testB | val | test |
| *LVLM-based method* | | | | | | | | | |
| ○ LLaVA 1.5 (Liu et al., 2023) | 7B | 5.8 | 6.5 | 5.1 | 5.4 | 6.0 | 4.8 | 7.9 | 8.1 |
| ○ LLaVA-Next (Liu et al., 2024) | 7B | 10.5 | 11.2 | 9.8 | 9.9 | 10.6 | 9.1 | 11.5 | 11.8 |
| ○ InternVL-2.5 (Chen et al., 2024) | 7B | 12.6 | 13.4 | 11.9 | 12.0 | 12.8 | 11.2 | 13.8 | 14.0 |
| ○ InternVL-3 (Chen et al., 2024) | 8B | 13.1 | 14.0 | 12.5 | 12.5 | 13.3 | 11.8 | 14.5 | 14.8 |
| ○ InternVL-3 (Chen et al., 2024) | 8B | 13.7 | 14.5 | 13.0 | 13.1 | 14.0 | 12.4 | 15.1 | 15.4 |
| ○ InternVL-3 (Chen et al., 2024) | 9B | 14.2 | 15.1 | 13.6 | 13.7 | 14.6 | 13.0 | 15.8 | 16.1 |
| ○ Qwen2-VL (Wang et al., 2024) | 2B | 13.8 | 14.6 | 13.2 | 13.1 | 13.9 | 12.5 | 15.0 | 15.3 |
| ○ Qwen2-VL (Wang et al., 2024) | 7B | 15.8 | 16.7 | 15.2 | 15.3 | 16.2 | 14.7 | 17.1 | 17.4 |
| ○ Qwen2.5-VL (Bai et al., 2025) | 3B | 13.3 | 14.1 | 12.8 | 12.7 | 13.5 | 12.1 | 14.7 | 15.0 |
| ○ Qwen2.5-VL (Bai et al., 2025) | 7B | 15.5 | 16.4 | 14.9 | 15.0 | 15.9 | 14.4 | 16.8 | 17.1 |
| ● LLaVA OneVision (Zhang et al., 2024) | 7B | 10.8 | 11.5 | 9.4 | 10.1 | 10.9 | 12.5 | 11.9 | 11.4 |
| ● **Diffusion Fine-Tuning** | **8B** | **21.8** | **22.5** | **20.9** | **21.1** | **21.9** | **20.2** | **22.8** | **23.2** |

Table 1: **16 Point polygon grounding results of RefCOCO-Polygon series.** ● represents the model that DFT is based on, while ○ stands for the remaining models. DFT is far better than that of the others.

| BBOX Grounding | RefCOCO | | | RefCOCO+ | | | RefCOCOg | |
|---|---|---|---|---|---|---|---|---|
| | val | testA | testB | val | testA | testB | val | test |
| *LVLM-based method* | | | | | | | | |
| ○ KOSMOS-2 | 52.3 | 57.4 | 47.3 | 45.5 | 50.7 | 42.2 | 60.6 | 61.6 |
| ○ Shikra 7B | 87.0 | 90.6 | 80.2 | 81.6 | 87.4 | 72.1 | 82.3 | 82.2 |
| ○ VisionLLM-H | 86.7 | 86.7 | - | - | - | - | - | - |
| ○ OFA-L | 80.0 | 83.7 | 76.4 | 68.3 | 76.0 | 61.8 | 67.6 | 67.6 |
| ○ InternVL-2 2B | 82.3 | 88.2 | 75.9 | 73.5 | 82.8 | 63.3 | 77.6 | 78.3 |
| ○ InternVL-2 8B | 87.1 | 91.1 | 80.7 | 79.8 | 87.9 | 71.4 | 82.7 | 82.7 |
| ○ Qwen2-VL 2B | 87.6 | 90.6 | 82.3 | 79.0 | 84.9 | 71.0 | 81.2 | 80.3 |
| ○ Qwen2-VL 7B | 91.7 | 93.6 | 87.3 | 85.8 | 90.5 | 79.5 | 87.3 | 87.8 |
| ○ Qwen2.5-VL 3B | 89.1 | 91.7 | 84.0 | 82.4 | 88.0 | 74.1 | 85.2 | 85.7 |
| ○ Qwen2.5-VL 7B | 90.0 | 92.5 | 85.4 | 84.2 | 89.1 | 76.9 | 87.2 | 87.2 |
| ● **DFT 8B** | **92.2** | **93.8** | **88.0** | **85.9** | **90.8** | **79.4** | **88.0** | **88.4** |

| 3D BBOX Grounding | Indoor 3D Datasets | | | | Outdoor | |
|---|---|---|---|---|---|---|
| | Objtro. | SUN-R. | ARKit. | Hyper. | KITTI | nuSc. |
| *LVLM-based method* | | | | | | |
| ○ LLaVA 1.5 7B | 40.7 | 23.1 | 1.1 | 4.2 | 2.3 | 1.4 |
| ○ LLaVA-Next 7B | 41.0 | 23.5 | 1.2 | 4.4 | 2.4 | 1.5 |
| ○ InternVL-2.5 7B | 45.3 | 35.7 | 1.5 | 3.9 | 25.0 | 3.2 |
| ○ InternVL-3 8B | 47.0 | 37.0 | 1.8 | 4.5 | 27.3 | 4.0 |
| ○ InternVL-3 9B | 48.9 | 38.5 | 2.5 | 6.8 | 32.0 | 5.8 |
| ○ Qwen2-VL 2B | 44.4 | 36.3 | 2.1 | 4.6 | 36.1 | 4.5 |
| ○ Qwen2-VL 7B | 51.0 | 38.4 | 2.0 | 4.3 | 27.2 | 3.8 |
| ○ Qwen2.5-VL 3B | 14.5 | 39.0 | 1.2 | 3.7 | 6.6 | - |
| ○ Qwen2.5-VL 7B | 9.0 | 39.0 | 4.6 | 3.8 | 2.0 | - |
| ● LLaVA OV 7B | 43.2 | 25.0 | 1.5 | 5.1 | 2.9 | 2.0 |
| ● **DFT 8B** | **52.2** | **39.3** | **4.8** | **9.1** | **39.7** | **8.4** |

Table 2: **Bbox grounding on RefCOCO.**          Table 3: **3D BBox grounding results.**

**Polygon grounding.** To test DFT's advantage over autoregressive (AR) models in generating the long, precise coordinate sequences needed for polygon grounding, we used our custom RefCOCO-Polygon (16-point) dataset. As shown in Table 1, DFT's superiority is clear. On the RefCOCOg test set, our model's score of 22.8 surpasses the top-performing autoregressive model, Qwen2-VL (17.4), by 31.0%. This marks an even larger 61.6% performance gain over our architectural baseline, LLaVA OneVision, which scored 12.5. This substantial improvement validates our hypothesis that DFT's diffusion paradigm, through parallel iterative optimization, transforms localization into a robust optimization problem, overcoming the limitations of AR models with complex outputs.

**Monocular 3D Object Grounding.** To validate our DFT framework's generalization, we applied it to the challenging task of monocular 3D object grounding, which requires predicting an object's 9-DoF pose (center, dimensions, and orientation). As shown in Table 3, model performance fluctuates significantly across datasets like ARKitScenes, KITTI, and nuScenes, largely due to task difficulty and limited data in some cases (e.g., KITTI). Qwen2.5-VL's particularly poor performance on outdoor datasets is noteworthy. Full results are shown in Table 6. We suspect its training, overly focused on 2D images, has limited its capability in scenes requiring long-range depth perception. Despite these challenges, our DFT model performed strongly on all benchmarks. Its key advantage is its iterative refinement capability: as shown in Appendix F, DFT progressively corrects even significant initial errors in the 3D box during its denoising steps to achieve a precise final prediction. This confirms that DFT's iterative optimization is a general framework for generating structured spatial information, not just 2D coordinates.

Table 4: **Ablation Study of coordinate representation (floating-point vs. integer) and polygon complexity (vertex count) on the RefCOCO-Polygon dataset.**

| Coordinate Type | Point Number | RefCOCO-Polygon | | |
|---|---|---|---|---|
| | | val | testA | testB |
| 0.00 - 0.99 | 16 | 16.5 | 18.1 | 17.4 |
| 0.000 - 0.999 | 16 | 17.3 | 20.0 | 17.6 |
| Normed 000-999 | 8 | **23.4** | **25.5** | **23.9** |
| Normed 000-999 | 16 | **21.5** | **22.1** | **20.5** |
| Normed 000-999 | 24 | 4.5 | 4.6 | 3.4 |

Table 5: **Ablation study on 3D parameter normalization.** The results clearly highlight the critical impact of depth (z) normalization, especially for outdoor scenes such as KITTI.

| 3D Normlization Setting | Indoor SUN-R | Outdoor KITTI |
|---|---|---|
| Baseline: All Normed | **39.3** | **39.7** |
| Without xy normalization | 33.1 | 34.5 |
| Without z normalization | 23.8 | 25.2 |
| Without dimension normalization | 34.0 | 35.9 |
| Without rotation angle normalization | 35.5 | 33.1 |

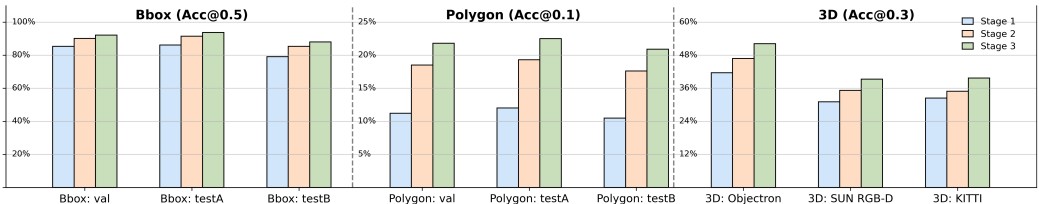

Figure 6: **Ablation Study on the Effectiveness of Hierarchical Curriculum Learning (HCL).** Performance gains across the three HCL stages (from coarse to fine). The strategy is particularly effective for complex tasks, with the most dramatic improvements seen in polygon grounding.

### 4.3 ABLATION STUDY

**Ablation on Hierarchical Curriculum Learning.** To validate our Hierarchical Curriculum Learning (HCL) strategy, we analyzed its performance across each learning stage. Figure 6 reveals that the greatest performance improvements occurred in the most complex tasks, with polygon grounding showing a substantial gain, followed by the considerable growth in the 9-DoF 3D grounding task. Conversely, the simple bounding box task saw the smallest increase, indicating that the more complex a target's representation (e.g., polygon vertices or 3D parameters), the more it benefits from the coarse-to-fine planning of our HCL strategy. This demonstrates our strategy's effectiveness.

**Analysis of Coordinate Representation and Polygon Complexity.** We first tested the impact of different coordinate representations (at a fixed 16 vertices). As shown in Table 4, our normalized integer representation ('000-999') significantly outperformed floating-point formats, likely because its integer range offers a clearer perceptual scale for learning spatial relationships. A higher vertex count exponentially increases task difficulty, providing a more rigorous test of a model's global planning and robustness. For this reason, we chose 16-point grounding as our core benchmark, which is challenging enough to highlight our DFT model's superior stability.

**Analysis of 3D Normalization Representation.** Our ablation analysis on 3D parameter normalization revealed a clear hierarchy of importance as shown in Table 5. Depth (z) normalization is the most critical factor, as its removal caused performance to drop sharply by 15.5 (SUN-R) and 14.5 (KITTI) points. While the 2D center point (xy) is the second most important parameter and dimension/rotation are the least, our experiments conclusively show that accurate depth estimation is the single most critical factor for monocular 3D detection.

## 5 CONCLUSION

We introduce diffusion fine-tuning (DFT) to improve the localization accuracy of LVLMs. It avoids the error accumulation and lack of global planning inherent in AR models that generate vertices sequentially. DFT reformulates visual localization as a coarse-to-fine optimization problem. This is enabled by our Hierarchical Coordinate Decomposition (HCD), which splits coordinates into "hundreds, tens, units" levels. We train this with Hierarchical Curriculum Learning (HCL), an easy-to-hard staged curriculum that ensures both training stability and high precision. DFT sets a new state-of-the-art on our RefCOCO-Polygon benchmark and for 2D bounding boxes on RefCOCO, and also generalizes effectively to the challenging 9-DoF 3D localization task. Please note that limitations including failure cases and LLM statements, are in Appendix C.

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

# A    PROOFS OF THEOREMS AND LEMMAS

## A.1    PRELIMINARY AND NOTATION

**Diffusion Models** are latent variable-based generative models with two core Markov chains: a forward noising process and a reverse denoising process. **The fixed forward process** injects Gaussian noise into data samples $\mathbf{x}_0 \sim p(\mathbf{x}_0)$ over $T$ steps, with each step defined as $q(\mathbf{x}_t|\mathbf{x}_{t-1}) = \mathcal{N}(\mathbf{x}_t; \sqrt{1-\beta_t}\mathbf{x}_{t-1}, \beta_t\mathbf{I})$, where $\beta_t \in (0,1)$ is a predefined hyperparameter controlling the noise magnitude at each step. eventually transforming $\mathbf{x}_0$ into near-standard normal noise $\mathbf{x}_T \sim \mathcal{N}(\mathbf{0}, \mathbf{I})$. The joint forward probability is $q(\mathbf{x}_{1:T}|\mathbf{x}_0) = \prod_{t=1}^{T} q(\mathbf{x}_t|\mathbf{x}_{t-1})$. **The learnable reverse process**, via a neural network ($\theta$), reverses this by denoising $\mathbf{x}_T$ to generate samples matching the original distribution, with joint probability $p_\theta(\mathbf{x}_{0:T}) = p(\mathbf{x}_T)\prod_{t=1}^{T} p_\theta(\mathbf{x}_{t-1}|\mathbf{x}_t)$. Training optimizes $\theta$ by minimizing the ELBO loss $\mathcal{L}_{\text{VLB}}$, which combines a reconstruction term, denoising matching term, and prior matching term to ensure proper denoising and distribution alignment.

**For discrete data like text**, we use a specialized discrete denoising diffusion model, where data (e.g., tokens) are represented as one-hot vectors $\mathbf{x}_t \in \{0,1\}^K$ ($K$ = vocabulary size). **The forward noising process** describes transitions from $t-1$ to $t$ via a categorical distribution: $q(\mathbf{x}_t|\mathbf{x}_{t-1}) = \text{Cat}(\mathbf{x}_t; \mathbf{Q}_t^\top \mathbf{x}_{t-1})$, with $\mathbf{Q}_t \in [0,1]^{K \times K}$ (a transition matrix, $[\mathbf{Q}_t]_{ij}$ = probability of state $i \rightarrow j$). We adopt 'Absorbing Discrete Diffusion', where $\mathbf{Q}_t = (1-\beta_t)\mathbf{I} + \beta_t\mathbf{1m}^\top$ ($\beta_t$ = noise scalar, $\mathbf{I}$ = identity matrix, $\mathbf{m}$ = one-hot `[MASK]` (absorbing state), $\mathbf{1}$ = all-ones column vector) — each token stays unchanged with probability $1-\beta_t$ or becomes `[MASK]` with $\beta_t$. By multiplying single-step matrices, we compute the distribution from $\mathbf{x}_0$ to $\mathbf{x}_t$ directly: let $\alpha_t = \prod_{i=1}^{t}(1-\beta_i)$, the cumulative matrix $\overline{\mathbf{Q}}_t = \alpha_t\mathbf{I} + (1-\alpha_t)\mathbf{1m}^\top$, so

$$q(\mathbf{x}_t|\mathbf{x}_0) = \text{Cat}(\mathbf{x}_t; \overline{\mathbf{Q}}_t^\top \mathbf{x}_0) = \alpha_t\mathbf{x}_0 + (1-\alpha_t)\mathbf{m}, \tag{7}$$

As $t$ grows, $\alpha_t \rightarrow 0$, and $\mathbf{x}_T$ nearly becomes random noise. To address the discrete-time framework's ($t \in [0,T]$) fixed noise limitation, we extend to continuous time ($t \in [0,1]$, $T \rightarrow \infty$), defining the forward process as $q(\mathbf{x}_t|\mathbf{x}_s)$ ($0 \leq s < t \leq 1$) — this formulation is used subsequently.

## A.2    CONTINUOUS-TIME DISCRETE DIFFUSION PROCESSES

**Continuous-Time State Transition.** Following Eq. 7 and $q(\boldsymbol{x}_t|\boldsymbol{x}_0) = \sum_{\boldsymbol{x}_s} q(\boldsymbol{x}_t|\boldsymbol{x}_s)q(\boldsymbol{x}_s|\boldsymbol{x}_0)$, the forward process from any earlier noisy state $\mathbf{x}_s$ to a later noisy state $\mathbf{x}_t$ can be precisely described as follows:

$$q(\boldsymbol{x}_t|\boldsymbol{x}_s) = \text{Cat}(\boldsymbol{x}_t; \overline{\boldsymbol{Q}}_{s|t}^\top \boldsymbol{x}_s) = \frac{\alpha_t}{\alpha_s}\boldsymbol{x}_s + (1 - \frac{\alpha_t}{\alpha_s})\boldsymbol{m}, \tag{8}$$

This indicates that from $s$ to $t$, an unmasked token has a probability of $\alpha_t/\alpha_s$ to remain unchanged, and a probability of $1 - \alpha_t/\alpha_s$ to be masked.

**True Posterior.** Similarly, given the original data $\mathbf{x}_0$ and the noisy data $\mathbf{x}_t$, we can derive the *true posterior distribution* $q(\mathbf{x}_s|\mathbf{x}_t, \mathbf{x}_0)$ for reversing from $t$ back to $s$. This distribution is the target that our model needs to learn:

$$q(\boldsymbol{x}_s|\boldsymbol{x}_t, \boldsymbol{x}_0) = \frac{q(\boldsymbol{x}_t|\boldsymbol{x}_s)q(\boldsymbol{x}_s|\boldsymbol{x}_0)}{q(\boldsymbol{x}_t|\boldsymbol{x}_0)} = \begin{cases} \frac{\alpha_s - \alpha_t}{1-\alpha_t}\boldsymbol{x}_0 + \frac{1-\alpha_s}{1-\alpha_t}\boldsymbol{m} & \text{if } \boldsymbol{x}_t = \boldsymbol{m}, \\ \boldsymbol{x}_0 & \text{if } \boldsymbol{x}_t \neq \boldsymbol{m}. \end{cases} \tag{9}$$

1) if a token at time $t$ *is not the mask*, then it must be the original token $\mathbf{x}_0$ at the earlier time $s$. 2) If a token at time $t$ *is the mask*, then its distribution at time $s$ is a linear combination of $\mathbf{x}_0$ and $\mathbf{m}$.

**Modeling and Training Objective.** Our goal is to train a **denoising model** $p_\theta$ to approximate the aforementioned true posterior distribution. The core of this model is a neural network (e.g., a Transformer), which we denote as $f_\theta$. Its task is to take a noisy sequence $\mathbf{x}_t$ as input and predict the most probable original sequence $\mathbf{x}_0$.

The training objective of the model is to minimize the KL divergence between $p_\theta(\mathbf{x}_s|\mathbf{x}_t)$ and $q(\mathbf{x}_s|\mathbf{x}_t, \mathbf{x}_0)$. In the continuous-time limit ($T \to \infty$), the total ELBO loss can be simplified to an integral over the time interval $[0, 1]$:

$$\mathcal{L} = \int_0^1 w(t) \cdot \mathbb{E}_{q(\mathbf{x}_t|\mathbf{x}_0)} \left[ \mathrm{Loss}_t(\mathbf{x}_0, f_\theta(\mathbf{x}_t)) \right] dt, \tag{10}$$

where the loss term $\mathrm{Loss}_t$ is a cross-entropy loss calculated only on tokens that are masked, and $w(t)$ is a weight related to the noise schedule. We adopt a common linear noise schedule where $\alpha_t = 1 - t$, which simplifies the weight term $w(t)$ to $\frac{1}{t}$. This weight implies that the model will pay more attention to denoising tasks in the early stages (when $t$ is small).

**Final Loss Applied to Text Sequences.** In practice, we independently apply the aforementioned theory to a text sequence $\mathbf{x}_t = [\mathbf{x}_t^1, \dots, \mathbf{x}_t^N]$ containing $N$ tokens. Instead of computing the complete integral, we estimate it using the Monte Carlo method: in each training iteration, we randomly sample a time $t \sim U(0, 1)$ for each data point and calculate the loss at that moment. Ultimately, for a given time $t$, the loss function applied to the text sequence is:

$$\mathcal{L}_t = \frac{1}{t} \mathbb{E}_{q(\mathbf{x}_t|\mathbf{x}_0)} \left[ -\sum_{n=1}^N \delta_{\mathbf{x}_t^n, \mathbf{m}}(\mathbf{x}_0^n)^\top \log f_\theta(\mathbf{x}_t^{1:N})_n \right], \tag{11}$$

Here, $f_\theta(\mathbf{x}_t^{1:N})_n$ represents the output probability distribution at the $n$-th position obtained by feeding the entire noisy sequence into the Transformer model. $\delta_{\mathbf{x}_t^n, \mathbf{m}}$ is an indicator function that ensures the loss is calculated only on those tokens that are masked.

## B  Inference Pseudocode

---

**Algorithm 1** : Parallel Denoising Inference

---

**Input:** **I**mage $I$; **T**ext $T_{\text{ins}}$; **T**otal denoising steps $T_{\text{steps}}$; **N**umber of vertices for the polygon $N$.
**Output:** Final integer coordinate sequence $S_{\text{final}}$.

                                                         **Stage 1: Initialization**

1:   $t \leftarrow 1$
2: **for** $k \in \{H, T, U\}$ **do**
3:     $S_t^k \leftarrow$ an all-[MASK] sequence of length $2N$
4: **end for**

                                    **Stage 2. Parallel Iterative Denoising**

5: **for** $t = 1, 1 - \frac{1}{T_{\text{steps}}}, \dots, \frac{1}{T_{\text{steps}}}$ **do**
6:     $s \leftarrow t - \frac{1}{T_{\text{steps}}}$                 *// Predict clean sequence distributions in parallel*
7:     $(\hat{p}_0^H, \hat{p}_0^T, \hat{p}_0^U) \leftarrow f_\theta\left((S_t^H, S_t^T, S_t^U), t, I, T_{\text{instr}}\right)$
                                      *// Update noise states in parallel*
8:     **for** $k \in \{H, T, U\}$ **do**
9:        $\hat{S}_0^k \leftarrow \text{decode}(\hat{p}_0^k)$ {e.g., using greedy only, not nucleus sampling}
10:      $S_s^k \leftarrow \text{sample\_from\_posterior}(S_t^k, \hat{S}_0^k, t, s)$ {Based on $q(x_s|x_t, x_0)$}
11:    **end for**
                                                   *// Update joint state*
12:    $(S_t^H, S_t^T, S_t^U) \leftarrow (S_s^H, S_s^T, S_s^U)$
13: **end for**

                                      **Stage 3. Recomposition & Output**

14: Extract final clean sequence estimates $(\hat{S}_0^H, \hat{S}_0^T, \hat{S}_0^U)$
15: Initialize an empty sequence $S_{\text{final}}$
16: **for** $i = 1$ to $2N$ **do**
17:     $d^H \leftarrow \hat{S}_0^H[i]$, $d^T \leftarrow \hat{S}_0^T[i]$, $d^U \leftarrow \hat{S}_0^U[i]$, $c_i \leftarrow 100 \cdot d^H + 10 \cdot d^T + d^U$, Append $c_i$ to $S_{\text{final}}$
18: **end for**
19: **return** $S_{\text{final}}$

---

## C    Limitations and Statements

**Limitations.** Despite its strong performance, the DFT framework still has some limitations, as illustrated by the failure cases in Figure 14. Specifically, the model can suffer from polygon disappearance, where the outline fails to generate despite accurate positioning, and at times, it may only ground a portion of the target object rather than its entirety. In 3D grounding tasks, the model can lead to object misalignment or inaccurate spatial positioning. Furthermore, grounding failures can occur when multiple outdoor objects are in close proximity to each other. These issues indicate that the model's robustness requires further improvement, particularly in complex scenarios involving ambiguous visual information or highly overlapping and proximate objects.

**LLM Usage Statement.** During the preparation of this manuscript, we utilized a Large Language Model (LLM) as an assistive tool. The primary role of the LLM was for language polishing, which included improving grammar, enhancing the clarity and fluency of the text, and ensuring consistent use of technical terminology.

All core research ideas, methodologies, experimental designs, results, and conclusions were conceived and formulated exclusively by the human authors. The LLM did not contribute to any of the substantive scientific aspects of this work. We have carefully reviewed and edited all text modified with the assistance of the LLM and take full responsibility for the final content of the paper.

**Reproducibility Statement.** We have submitted parts of our core source code. This includes:

- The core script for inference and logging (`inference.py`), which supports multimodal inference on single/multiple images and video frames and logs outputs such as generated responses, bounding boxes, and camera intrinsics to JSON files for traceability.
- An evaluation toolkit for 2D polygon-based localization and segmentation tasks (`eval_poly_rec.py`), which parses polygon coordinates from the model's text output and computes key metrics such as polygon Intersection over Union (IoU) and accuracy.
- A utility for evaluating 9-Degree-of-Freedom (9-DoF) 3D bounding boxes (`utils_bbox3d_rec.py`). This tool is responsible for parsing 3D parameters from text outputs, converting them into 3D corner coordinates, and computing 3D IoU.

We also submitted the DFT demo videos, which include:

- `Indoor 3D Bounding Box Grounding.mp4`
- `Outdoor 3D Bounding Box Grounding.mp4`
- `polygon grounding.mp4`

## D    Ref3D Datasets

We introduce Ref3D, a large, diverse benchmark dataset tailored for training Large Multimodal Models (LMMs) in 3D Visual Grounding. The core objective of this task is to enable a model to locate and regress the 9-Degree-of-Freedom (9-DoF) 3D bounding box of a target object from a 2D image, guided by natural language. To achieve this, we consolidated and unified six authoritative 3D vision datasets covering indoor, outdoor, real-world, and synthetic scenes. A key design principle of Ref3D is the 'one image, one target' philosophy: each image is curated to link with a single target object and its text description. This one-to-one mapping aims to deliver clear, unambiguous training signals, letting the model precisely learn cross-modal alignment between language, imagery, and 3D space.

The core of the scheme is to quantize all 9-Degree-of-Freedom (9-DoF) parameters (covering location, dimension, and rotation) and map them to the integer range $[0, 999]$, enabling seamless integration into the generative framework of Large Multimodal Models (LMMs). Specific strategies for each parameter are as follows:

1. **2D Projected Coordinates** $(x, y)$
   Linear normalization is performed directly on the projected coordinates of the object's 3D center on the image plane.

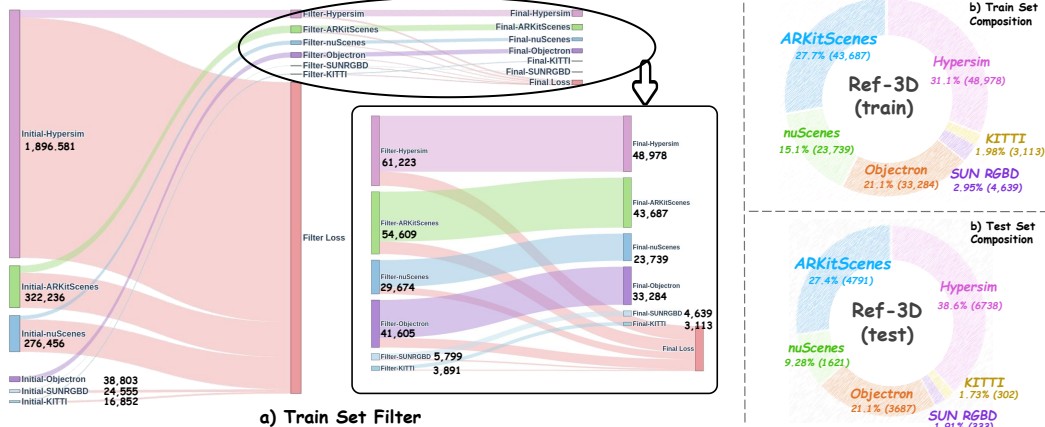

Figure 7: **Ref3D Dataset Filter Process and Final Composition.**

2. **Depth ($z$)**
   A two-step method of "logarithmic transformation + linear normalization" is adopted to address the long-tail distribution of depth in real-world scenes:

   - First, map the actual depth (approximately $0.018\,\text{m}$ to $148.4\,\text{m}$) to the logarithmic scale of $[-4.0, 5.0]$ via logarithmic transformation;
   - Then apply linear normalization to this logarithmic interval to map it to $[0, 999]$;
   - During denormalization, recover the actual depth by performing an exponential operation on the logarithmic scale value.

3. **Physical Dimensions (width $w$, height $h$, length $l$)**
   The dimensions are first linearly mapped to a predefined physical range (e.g., $[0, 15]$ meters), followed by linear normalization to $[0, 999]$.

4. **Rotation Angles (yaw $r_1$, pitch $r_2$, roll $r_3$)**
   The angles are first converted to radians to ensure coverage of the full angular range $[0, 2\pi]$, and then linearly normalized to $[0, 999]$.

To ensure each sample in the Ref3D dataset has a clear, unambiguous, and suitable target for visual grounding, we designed an automated data curation and annotation pipeline using the Large Multimodal Model Doubao-1.5-VL. This pipeline is engineered to intelligently select one optimal target for each image from its original multiple 3D annotations and generate a high-quality natural language description for it. Specifically, it operates in two stages:

1. **First stage (Candidate Filtering):** We feed the model an image and all its raw 3D annotations, prompting it to automatically discard heavily occluded, blurry, or truncated objects based on criteria of visibility, clarity, and integrity. This process yields a high-quality set of candidate objects.

2. **Second stage (Target Selection and Description Generation):** We present the image and the refined candidate set to the model again, instructing it to select the single best target based on principles of visual salience (e.g., centrality and size). Concurrently, the model generates a natural language description for the selected target, detailing its appearance attributes (e.g., color, shape) and spatial relationships (e.g., 'next to the window').

This sophisticated two-stage curation process not only enforces our 'one image, one target' design principle but also automatically enriches each sample with semantic annotations, significantly boosting the dataset's quality and the effectiveness of the task.

# E FULL 3D BBOX GROUNDING RESULTS

Due to greater scene complexity, the model performs better on indoor datasets than on outdoor ones. To ensure a precise and unbiased evaluation across the diverse Ref3D benchmark, we report performance for each sub-dataset individually in Table 6, rather than an overall average that could be skewed. Most critically, the experimental results clearly demonstrate the outstanding

| | LLaVA 1.5 7B | LLaVA Next 7B | InternVL2.5 7B | InternVL3 8B | InternVL3 9B | Qwen2VL 2B | Qwen2VL 7B | Qwen2.5VL 3B | Qwen2.5VL 7B | LLaVA-One Vision 7B | DFT 8B |
|---|---|---|---|---|---|---|---|---|---|---|---|
| **Objectron** | | | | | | | | | | | |
| Acc 0.1 | 66.7 | 67.1 | 72.1 | 73.5 | 74.8 | 72.6 | 77.3 | 54.1 | 42.9 | **68.5** → | **77.3** |
| Acc 0.3 | 40.7 | 41.0 | 45.3 | 47.0 | 48.9 | 44.4 | 51.0 | 14.5 | 9.0 | **43.2** → | **52.2** |
| Acc 0.5 | 14.8 | 15.2 | 21.0 | 23.1 | 26.0 | 19.3 | 25.7 | 0.0 | 0.0 | **16.9** → | **28.5** |
| **SUN RGB-D** | | | | | | | | | | | |
| Acc 0.1 | 45.1 | 45.5 | 61.5 | 62.8 | 64.1 | 61.9 | 65.0 | 60.7 | 65.2 | **47.1** → | **65.5** |
| Acc 0.3 | 23.1 | 23.5 | 35.7 | 37.0 | 38.5 | 36.3 | 38.4 | 39.0 | 39.0 | **25.0** → | **39.3** |
| Acc 0.5 | 3.0 | 3.2 | 16.2 | 17.5 | 19.0 | 16.8 | 19.2 | 12.9 | 14.4 | **4.1** → | **19.5** |
| **ARKitScenes** | | | | | | | | | | | |
| Acc 0.1 | 4.2 | 4.3 | 4.8 | 5.0 | 5.2 | 6.9 | 6.0 | 5.7 | 6.8 | **4.8** → | **7.5** |
| Acc 0.3 | 1.1 | 1.2 | 1.5 | 1.8 | 2.5 | 2.1 | 2.0 | 1.2 | 4.6 | **1.5** → | **4.8** |
| Acc 0.5 | 0.0 | 0.1 | 0.0 | 0.2 | 0.6 | 0.0 | 0.0 | 0.0 | 0.0 | **0.2** → | **1.0** |
| **Hypersim** | | | | | | | | | | | |
| Acc 0.1 | 13.4 | 13.6 | 15.2 | 16.1 | 18.5 | 18.5 | 16.7 | 17.1 | 16.1 | **14.9** → | **22.9** |
| Acc 0.3 | 4.2 | 4.4 | 3.9 | 4.5 | 6.8 | 4.6 | 4.3 | 3.7 | 3.8 | **5.1** → | **9.1** |
| Acc 0.5 | 1.2 | 1.3 | 0.9 | 1.2 | 1.7 | 1.1 | 1.0 | 0.0 | 0.0 | **1.6** → | **2.3** |
| **KITTI** | | | | | | | | | | | |
| Acc 0.1 | 7.0 | 7.2 | 48.1 | 51.5 | 65.3 | 62.6 | 50.7 | 14.2 | 7.0 | **8.2** → | **74.8** |
| Acc 0.3 | 2.3 | 2.4 | 25.0 | 27.3 | 32.0 | 36.1 | 27.2 | 6.6 | 2.0 | **2.9** → | **39.7** |
| Acc 0.5 | - | - | 7.6 | 8.9 | 15.4 | 18.5 | 8.6 | 2.0 | 0.0 | **0.5** → | **22.2** |
| **nuScenes** | | | | | | | | | | | |
| Acc 0.1 | 7.0 | 7.2 | 48.1 | 51.5 | 65.3 | 62.6 | 50.7 | 14.2 | 7.0 | **8.2** → | **74.8** |
| Acc 0.3 | 2.3 | 2.4 | 25.0 | 27.3 | 32.0 | 40.1 | 27.2 | 6.6 | 2.0 | **2.9** → | **39.7** |
| Acc 0.5 | - | - | 7.6 | 8.9 | 15.4 | 18.5 | 8.6 | 2.0 | 0.0 | **0.5** → | **22.2** |
| **Overall** | | | | | | | | | | | |
| Acc 0.1 | 21.9 | 22.6 | 25.9 | 26.9 | 28.4 | 27.7 | 27.4 | 21.1 | 18.7 | **24.2** → | **31.3** |
| Acc 0.3 | 11.1 | 11.8 | 13.1 | 14.2 | 15.8 | 13.5 | 14.6 | 5.7 | 4.7 | 12.9 → | 18.2 |
| Acc 0.5 | 3.7 | 3.9 | 6.0 | 6.7 | 7.5 | 5.5 | 6.6 | 0.0 | 0.0 | **4.3** → | **8.8** |

Table 6: **Full 3D Bbox Grounding Results.**

performance of our DFT method. Through a direct comparison with our base model, LLaVA-OneVision LLaVAOneVision, DFT endows it with exceptionally strong 3D grounding capabilities. For instance, on the Objectron dataset, DFT boosts the Acc@0.3 accuracy from 43.2 to 52.2; on the more challenging KITTI dataset, this improvement is particularly staggering, with the accuracy leaping from 2.9 to 39.7. This provides strong evidence that our method can effectively unlock and enhance the base model's potential in complex 3D spatial understanding and localization.

# F MORE QUANTITATIVE RESULTS AND FAILURE CASES

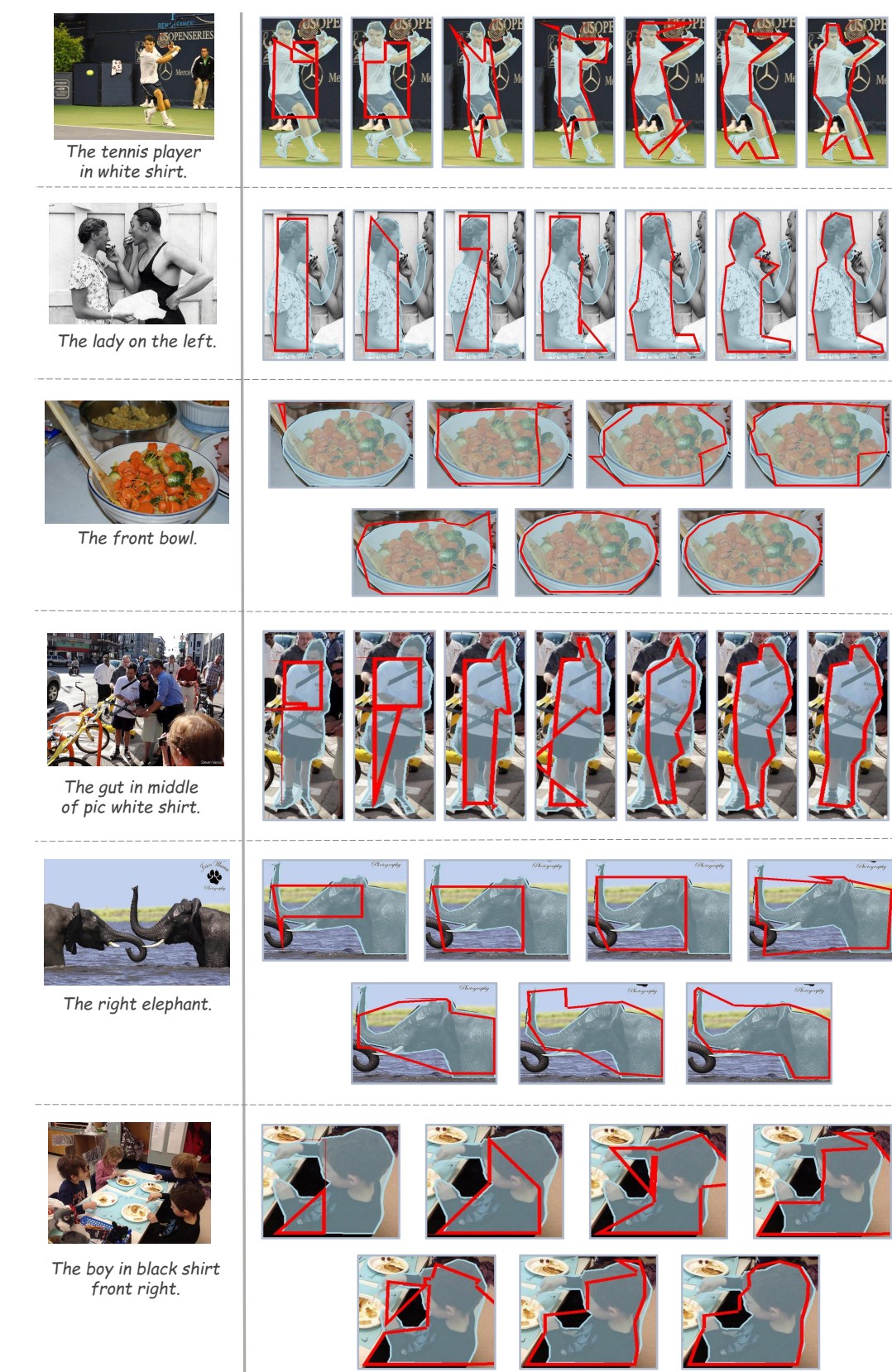

Figure 8: **16 Point polygon grounding demos.** There is no cherry pick here.

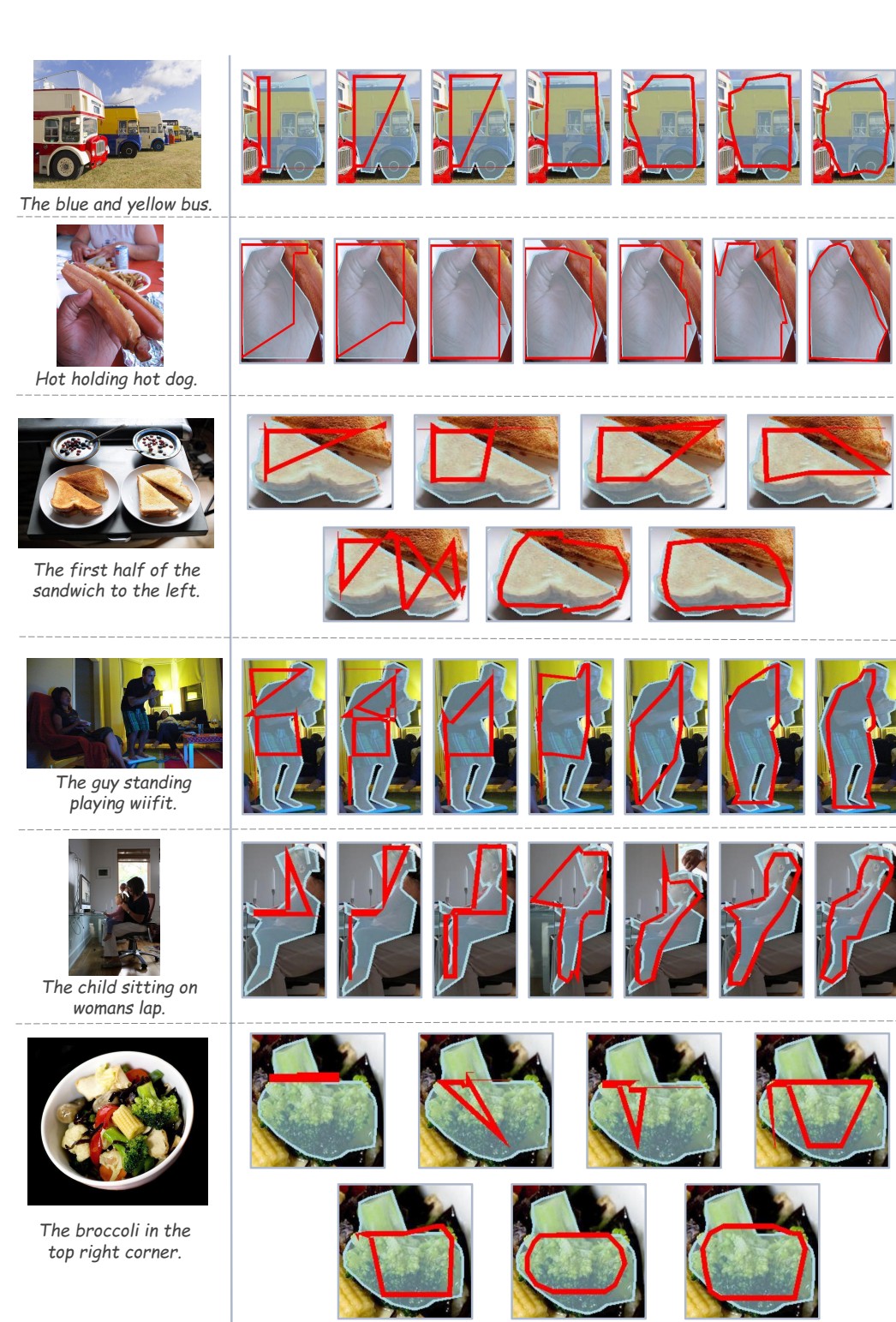

Figure 9: **16 Point polygon grounding demos.** There is no cherry pick here.

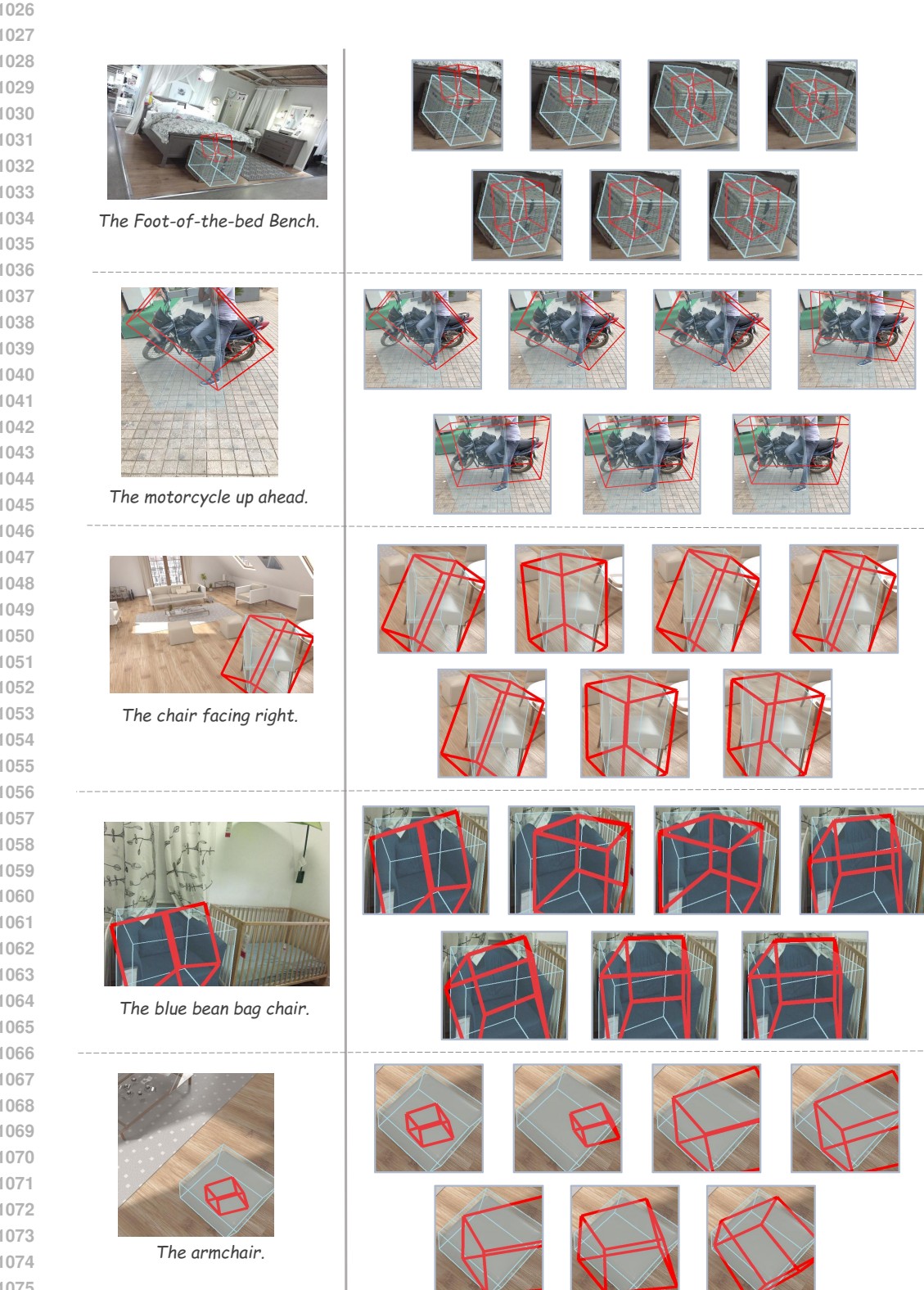

Figure 10: **Indoor 3D bounding boxes grounding demos.** From Objectron, SUN RGB-D, ARKitScenes, Hypersim, there is no cherry pick here.

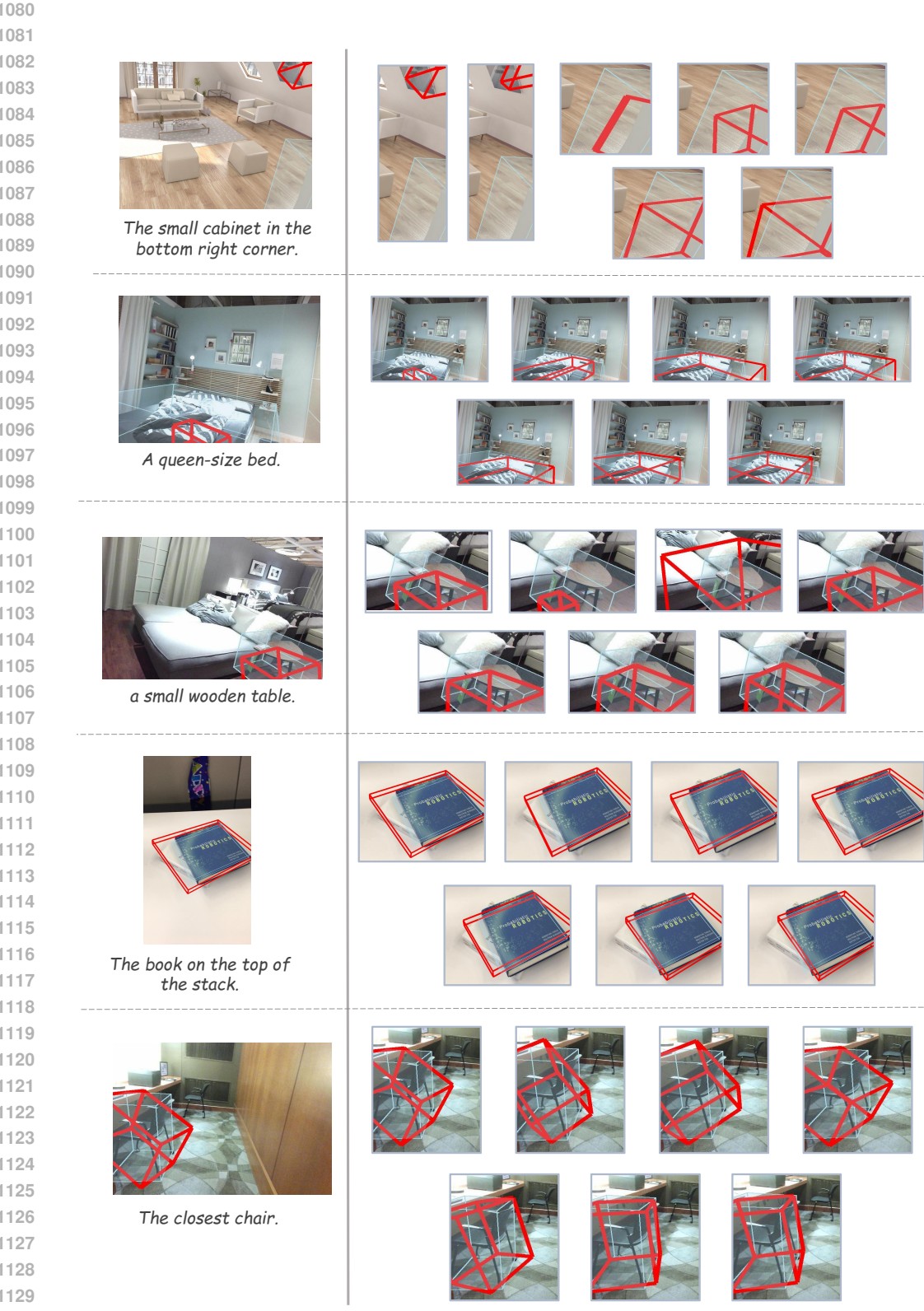

Figure 11: **Indoor 3D bounding boxes grounding demos.** From Objectron, SUN RGB-D, ARKitScenes, Hypersim, there is no cherry pick here.

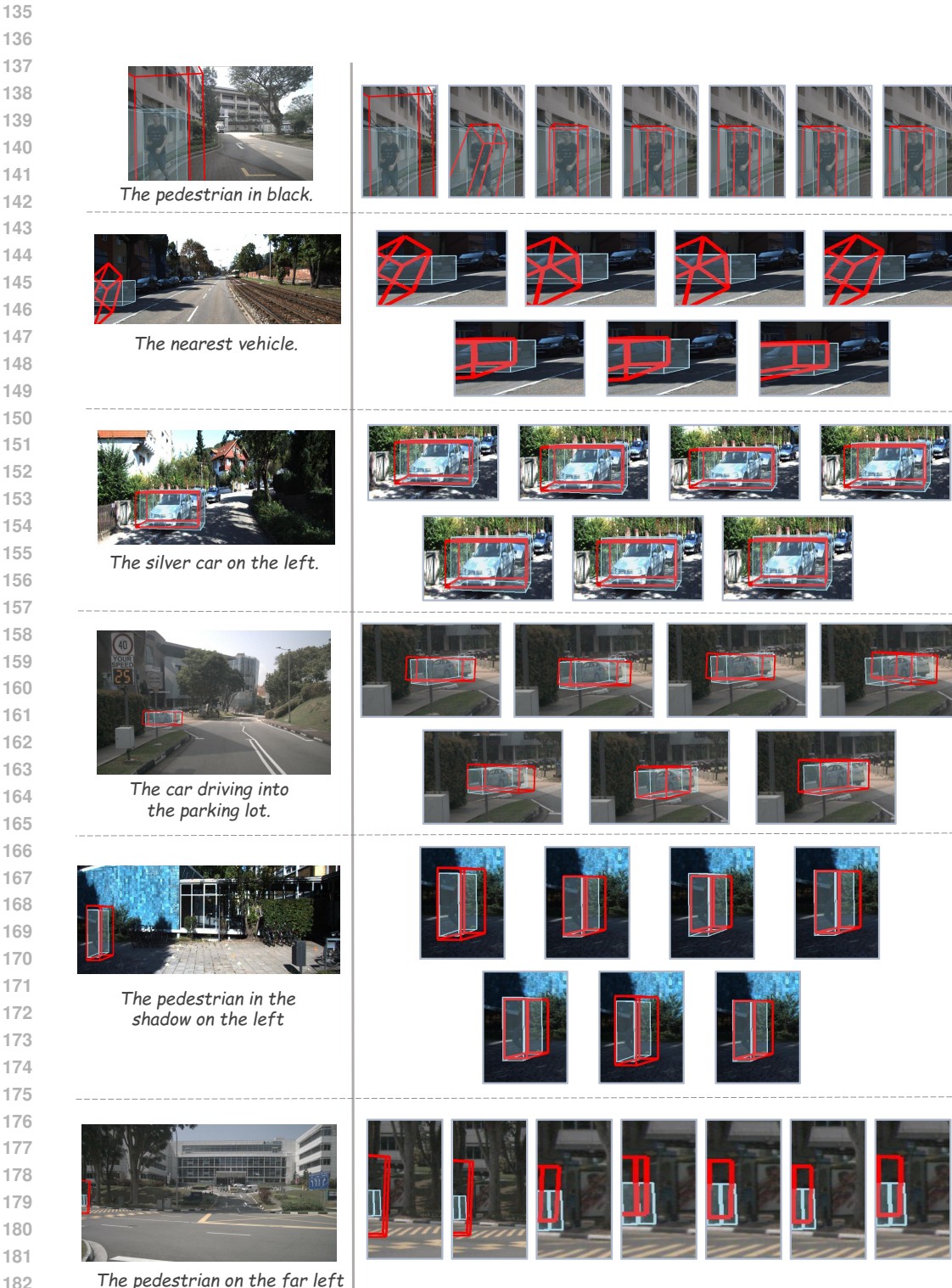

Figure 12: **Outdoor 3D bounding boxes grounding demos.** From KITTI & nuScenes, there is no cherry pick here.

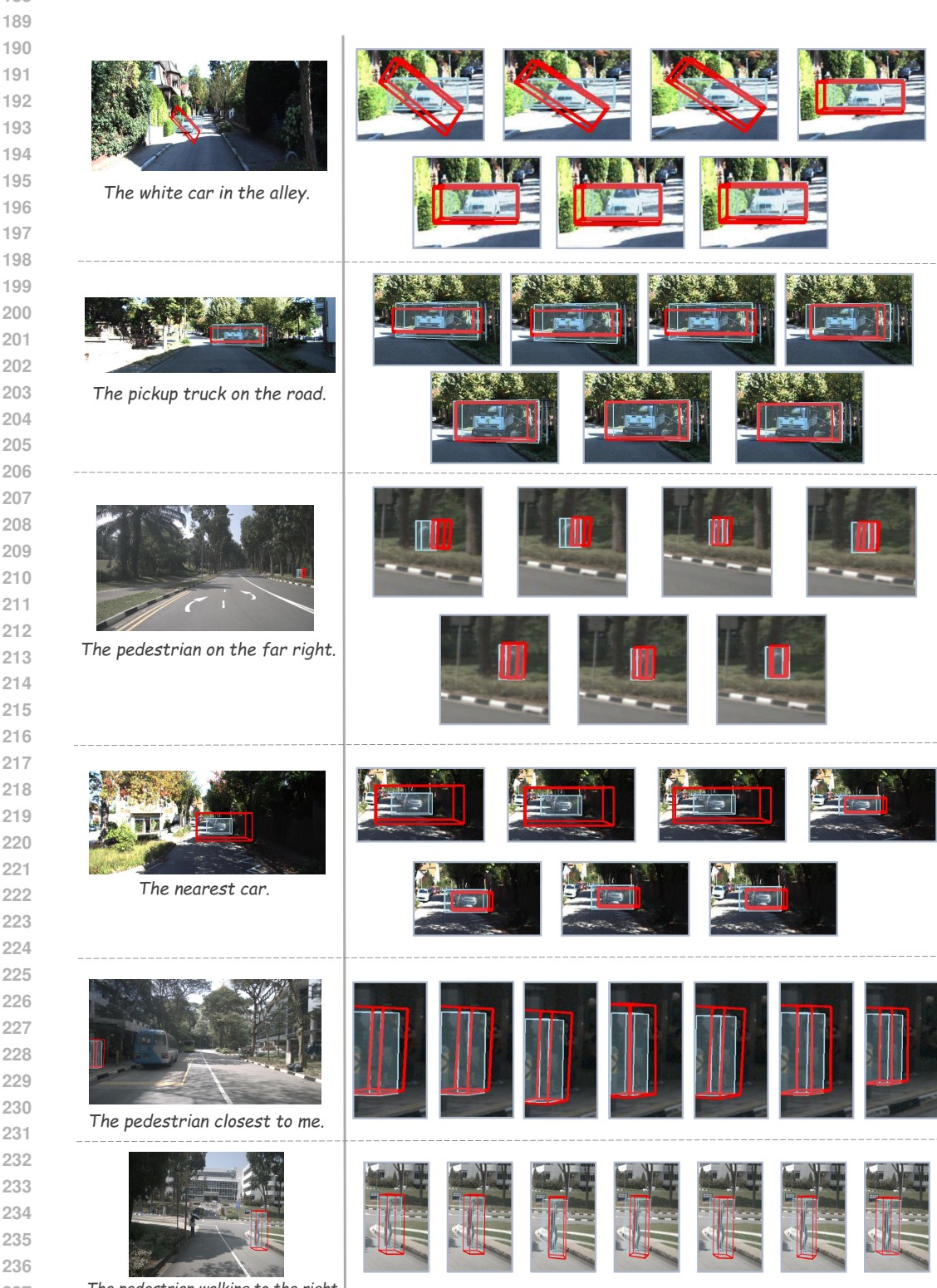

Figure 13: **Outdoor 3D bounding boxes grounding demos.** From KITTI & nuScenes, there is no cherry pick here.

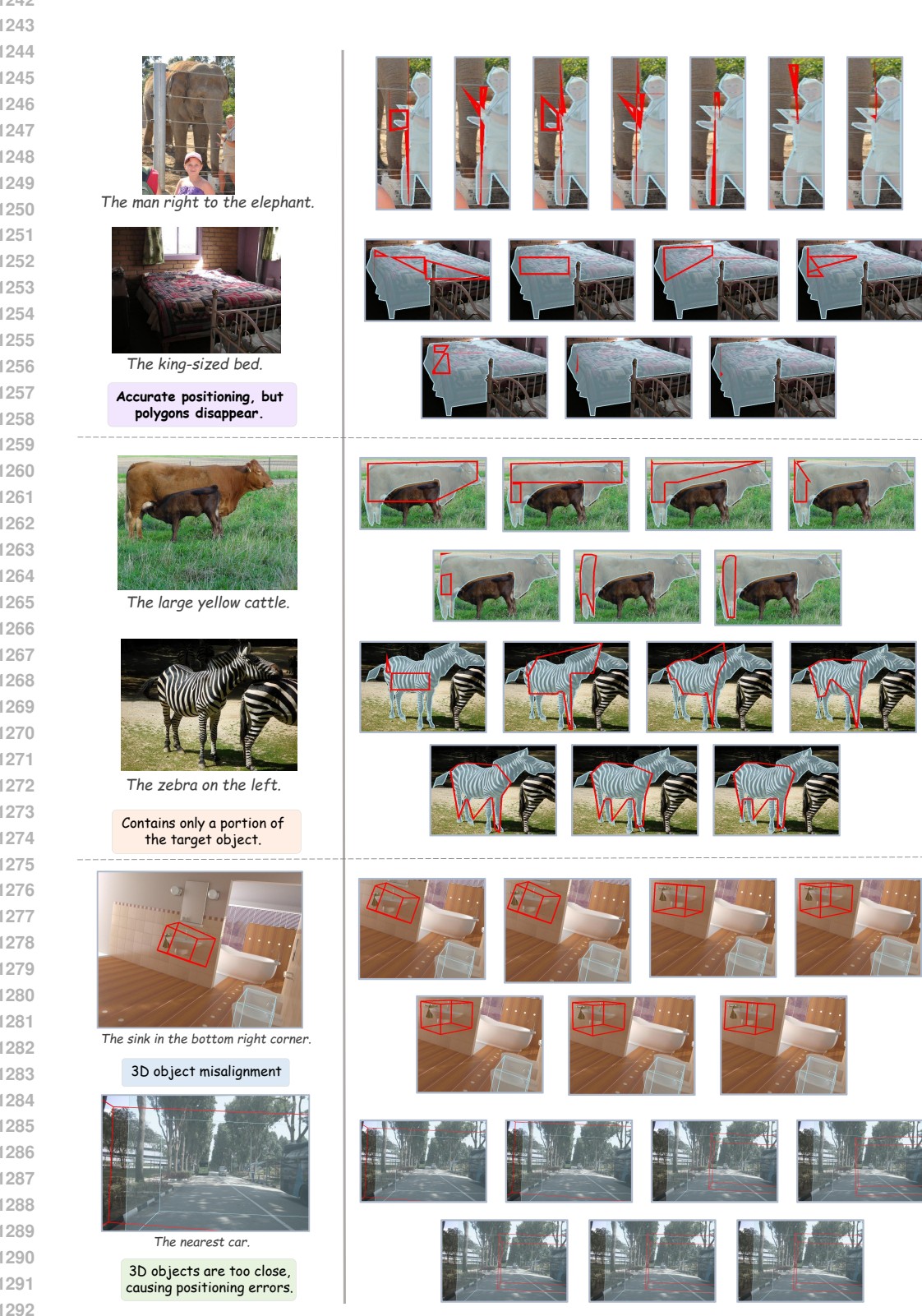

Figure 14: **DFT Failure cases.**

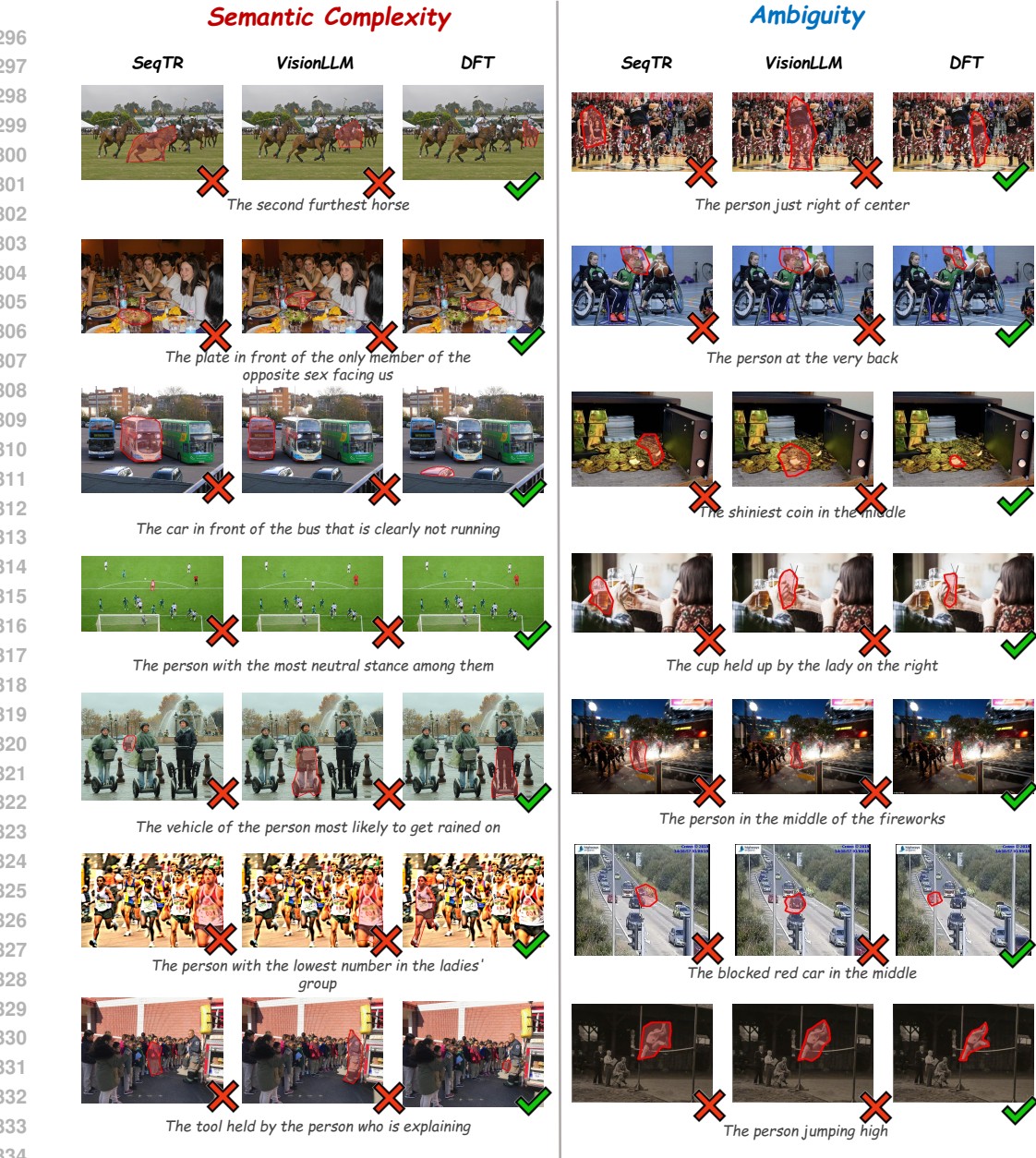

Figure 15: **Semantic Complexity and Ambiguity compared with other specific models and AR LVLMs.**

## G  REBUTTAL: SUPERIOR HANDLING OF COMPLEX SEMANTICS & OCCLUSION

We compared DFT against SeqTR (Specialist) and VisionLLM (AR Generalist) on geometric and semantic limit cases. For semantic complexity (Figure 15), SeqTR struggles with logic and Vision-LLM is prone to hallucination, whereas DFT succeeds by fusing LLM reasoning with Diffusion (e.g., grounding "shiniest coin"). regarding heavy occlusion, baselines miss hidden objects, but DFT's Global Planning infers holistic structure for accurate segmentation. Overall, DFT surpasses baselines in reasoning, robustness, and scalability.

