# OpenReview forum: "Diffusion Fine-Tuning: Iterative Refinement for Advanced Grounding with Diffusion Large Language Models"
_ICLR.cc/2026/Conference — ICLR 2026 Conference Desk Rejected Submission_

### Official Review · Reviewer_yKMe · 2025-10-30

**Soundness:** 2
**Presentation:** 3
**Contribution:** 2
**Rating:** 4
**Confidence:** 5

**Summary:**

The authors argue that autoregressive (AR) approaches to grounding are prone to error accumulation and cannot be optimized in parallel. Therefore, they propose a diffusion-like denoising method that generates object vertices in parallel for more effective grounding.

**Strengths:**

1. The authors' HCD and HCL designs are highly reasonable, effectively reducing regression difficulty and improving the stability of predicted contours.
2. DFT achieves strong performance on both 2D and 3D grounding, significantly outperforming previous vision-language models.

**Weaknesses:**

1. The authors' motivation and proposed concept are extremely similar to those of PVD [1]; the only difference is that the authors focus on vision-language models (VLMs). However, they overlook a direct comparison with this closely related approach. Moreover, the paper fails to cite several highly relevant prior works in the traditional visual grounding literature, such as SeqTR [2], PVD [1], Pix2Seq [3], VisionLLM [4].

2. Furthermore, the paper presents an ambiguous model positioning. If the model is intended to be general-purpose, the authors do not demonstrate its general multimodal understanding capability when performing polygon-based grounding. If it is a specialized model, they omit comparisons with existing dedicated 2D or 3D grounding models.

3. Given the method's complexity, the authors do not provide a comparison of inference speed against conventional VLMs for grounding tasks.

[1] Parallel Vertex Diffusion for Unified Visual Grounding
[2] SeqTR: A Simple yet Universal Network for Visual Grounding
[3] A Language Modeling Framework for Object Detection
[4] VisionLLM: Large Language Model is also an Open-Ended Decoder for Vision-Centric Tasks

**Questions:**

1. I believe that although the proposed model utilizes a vision-language model (VLM), it functions more like a specialized grounding model; therefore, the authors should include comparisons with existing specialized grounding models.
2. The authors need to discuss how their approach differs from conceptually similar works such as PVD, SeqTR, Pix2Seq, and VisionLLM.
3. The authors should analyze the inference overhead of their current approach to evaluate the cost-effectiveness of the performance gains.

---

> ### Author Response · Authors · 2025-11-25
> **Reply to Reviewer yKMe -- 1**
>
> # `Weakness 1 & Question 1: Comparision with specialist models and AR VLM`
> # `Weakness 2 & Question 2: DFT's Understanding Ability`
>
> We thank the reviewer for these references and will discuss them in the revision. Below, we differentiate DFT from these works by categorizing them into Specialist Models and Autoregressive LLMs.
>
> ### 💉 ***1. Comparison with Specialist Models (PVD, Pix2Seq, SeqTR)***
>
> **Category:** These methods are **Task-Specific Specialists** designed purely for geometric localization, lacking the general-purpose reasoning of LVLMs.
>
>
> * **Reasoning Capability:** Unlike specialists, DFT utilizes LVLM world knowledge to resolve complex semantic ambiguities (e.g., *'The first half of the sandwich'*). **We have included additional qualitative experiments in the supplementary material**, visually demonstrating DFT's superior capability in understanding and grounding complex semantic information where purely geometric specialist models fail.
>
> * **Reproducibility & Open Source:** Crucially, PVD is not open-sourced, limiting reproducibility. In contrast, we have submitted our code and are committed to full open-sourcing to advance the field.
>
> * **Parallel vs. Sequential:** While pioneering, these AR models suffer from `irreversible error accumulation`. DFT employs Parallel Global Optimization, refining all vertices simultaneously to solve this sequential bottleneck.
>
> * **DFT's 3D Superiority:**
>     Unifying 2D and 3D grounding, DFT demonstrates `exceptional depth prediction`. It successfully models complex 2D-to-3D mappings, a capability unmatched by segmentation models or 2D-centric AR VLMs.
>
> ---
>
> ### 🏗️ ***2. Comparison with VisionLLM (Autoregressive LVLM)***
>
> **Category:** VisionLLM serves as the Generalist LVLM baseline. Like our method, it aims to unify diverse vision tasks within a language model.
>
> * **The Paradigm Shift (Diffusion vs. AR):**
> The core difference is the generative paradigm. VisionLLM uses AR next-token prediction, which limits global planning. In contrast, DFT introduces Diffusion Fine-Tuning to treat outputs as holistic structures, yielding significant gains over AR baselines on complex spatial tasks (Table 1-3).
>
> * **General Multimodal Understanding:**
>     Beyond grounding, a true Generalist must possess strong multimodal comprehension. We compared DFT against VisionLLM on the **MVBench** benchmark to evaluate general perception and reasoning capabilities.
>
>
> > **Table 🔴: Comparison of General Video Understanding (MVBench)**
>
> | Model | Action Loc. | Object Exist. | Counterfactual Inf. | **Average** |
> | :--- | :---: | :---: | :---: | :---: |
> | **VisionLLM** | 57.4 | 82.5 | 56.5 | 58.1 |
> | **Intern-VL3 (8B)** | 60.4 | 81.7 | 58.3 | 61.5 |
> | **Qwen2.5-VL (7B)** | 59.2 | 82.7 | 58.9 | 60.9 |
> | **DFT (8B)** | **61.4** | **84.2** | **61.8** | **62.7** |
>
> > **Conclusion**:
> DFT achieves state-of-the-art performance on general video understanding, **surpassing not only the AR baseline (VisionLLM) but also top-tier SOTA models like Intern-VL3 (+1.2) and Qwen2.5-VL (+1.8)**. Notably, DFT shows superior capabilities in **Action Localization** and **Counterfactual Inference**, demonstrating that our discrete diffusion paradigm successfully empowers the model with robust spatiotemporal grounding and high-level reasoning capabilities, making it a highly versatile generalist.

---

> ### Author Response · Authors · 2025-11-25
> **Reply to Reviewer yKMe -- 2**
>
> # `Weakness 3 & Question 3: DFT's Inference Speed`
>
>
> ## 📦️ ***1. Theoretical Complexity Analysis***
>
>
> We analyze the fundamental efficiency difference by examining how frequently each model performs **Visual Interaction** (Cross-Attention between text and image).
>
> -----
>
> ###  **🅰️. AR Baseline: Decoupled Visual Encoding ($1 + N$)**
> Autoregressive models decouple visual processing from text generation.
>
>   * **Mechanism:** The heavy visual interaction occurs **only once** (during the prefill phase). The visual features are then stored in the KV-Cache.
>   * **Cost Formula:**
>
>     $$\text{Total Latency} \approx \underbrace{\text{Visual Interaction}}_{\text{Once (Heavy)}} + \sum\_{t=1}^{N} \text{Decoding Steps}$$
>   * **Result:** For a sequence of $N$ tokens, the model performs **1** visual interaction and **$N$** lightweight decoding steps.
>
> -----
> ### **🅱️. Diffusion Fine-Tuning (DFT): Coupled Visual Interaction ($T \times \text{Full}$)**
> In contrast, Diffusion models inherently couple visual interaction with the denoising process.
>
>   * **Mechanism:** At every denoising step, the entire noisy text sequence is updated. Therefore, the model must re-calculate the **Full Cross-Attention** between the text and visual features at every step.
>   * **Cost Formula:**
>     $$\text{Total Latency} \approx \sum_{t=1}^{T} (\underbrace{\text{Visual Interaction} + \text{Denoising}}_{\text{Heavy (Every Step)}})$$
>   * **Result:** The model performs **$T$** visual interactions. Since the per-step cost is high, we must reduce the total steps ($T$) to outperform AR.
>
> -----
>
> ## 🏗️ 2. Sensitivity Analysis: Why Multi-Token Decoding is Necessary
>
> Since DFT pays the `Visual Interaction Tax` at every step, we must reduce the total steps ($T$) to be competitive. We vary the **Effective Throughput** (tokens generated per visual interaction) to find the breakeven point.
>
> > **Table 🔴: Efficiency vs. Visual Interaction Frequency**
> > *Task: 16-Point Polygon ($N \approx 100$ tokens).*
>
>
> | Method | Visual Interactions ($T$) | Tokens/Step ($N/T$) | Latency (ms) $\downarrow$ | RefCOCO val |
> | :--- | :---: | :---: | :---: | :---: |
> | Qwen-2.5-VL (7B) | 1 | 1 | 700 | 15.5 |
> | | | | | |
> | DFT-8B | 100 | 1 | 6000 | 21.8 |
> | DFT-8B | 20 | 5 | 1400 | 23.0 |
> | DFT-8B | 10 | 10 | 900 | 20.1 |
> | DFT-8B | 6 | ~15 | 700 | 13.4 |
> | DFT-8B | 5 | 20 | 640 | 12.4 |
>
> -----
>
>
> ### **📊 Results Discussion**
>
> * **Peak Performance vs. Over-Refinement:** The AR baseline bottlenecks at **15.5**. DFT reaches peak accuracy at **$T=20$ (23.0)**. Notably, extending to $T=100$ slightly degrades performance (21.8), as excessive denoising steps lead to **geometric collapse** (e.g., shrinking or vanishing polygons).
> * **The Optimal Trade-off ($T=10$):** This is the efficiency sweet spot. We trade a **marginal 200ms latency increase** (900ms vs 700ms) for a **substantial accuracy boost (+4.6)** compared to the baseline.
> * **Efficiency Limit:** Strictly matching AR's speed ($T=6$, 700ms) compromises accuracy (13.4), confirming that $\sim10$ iterative visual interactions are essential for precise grounding.
>
> > **Conclusion:**
> While AR provides fixed low latency, it is limited in precision. At the optimal setting ($T=10$), DFT delivers **superior accuracy (+4.6)** at a **competitive speed**, establishing it as the better choice for high-precision tasks.

---

> ### Author Response · Authors · 2025-11-28
> **Looking forward to your reply**
>
> We hope that our revisions and explanations have adequately answered your questions. We have treated every comment with the utmost seriousness. We are eagerly hoping for a positive outcome, as this work is extremely significant to us.
>
>
>
> Since we have addressed the specific limitations you pointed out (e.g., adding the missing experiments), we hope you will **reconsider your assessment of our work**.

---

### Official Review · Reviewer_tJXR · 2025-11-01

**Soundness:** 3
**Presentation:** 3
**Contribution:** 2
**Rating:** 6
**Confidence:** 3

**Summary:**

This paper proposes a diffusion fine-tuning framework that reformulates visual grounding as a coarse-to-fine parallel global optimization process, effectively overcoming AR models’ irreversible error accumulation.

**Strengths:**

The paper is clearly written and the motivation from AR to diffusion is well explained and illustrated with intuitive figures. The technical formulation and execution are solid, and the experimental evaluation is thorough and convincing. Overall, the work demonstrates clear quality and significance for improving precise grounding performance.

**Weaknesses:**

1.The method may have non-trivial latency/compute overhead due to iterative diffusion + hierarchical refinement, but the paper does not quantify the speed–accuracy trade-off or compare runtime against strong AR baselines.

2.The method assumes a fixed 16-vertex polygon, and it is unclear how well it generalizes to extremely simple/complex shapes or variable-length polygons.

3.It is better to analyze limit cases (e.g., extremely complex shapes, ambiguous referring expressions, or heavy occlusion).

**Questions:**

1.I would like to see a concrete analysis of runtime / latency / compute cost, especially to quantify the diffusion vs AR speed–accuracy trade-off.

2.Although the method achieves SOTA compared to baselines, I would like to see more visual comparisons on difficult regimes (e.g., highly complex shapes, ambiguous text, heavy occlusion) to see the limit of AR and diffusion model respectively.

3.Since AR v.s. diffusion is a long-standing modeling choice trade-off, and this paper articulates clear advantages of diffusion for grounding, do you think this paradigm could generalize and surpass AR models in more general tasks beyond grounding? If yes, which categories do you believe are the most promising candidates (This is just a discussion, no additional experiments needed.)

---

> ### Author Response · Authors · 2025-11-25
> **Reply to Reviewer tJXR -- 1**
>
> # `Weakness 1 & Question 1: Time-consuming compared with the autoregressive models`
>
>
> ## 📦️ ***1. Theoretical Complexity Analysis***
>
>
> We analyze the fundamental efficiency difference by examining how frequently each model performs **Visual Interaction** (Cross-Attention between text and image).
>
> -----
>
> ###  **🅰️. AR Baseline: Decoupled Visual Encoding ($1 + N$)**
> Autoregressive models decouple visual processing from text generation.
>
>   * **Mechanism:** The heavy visual interaction occurs **only once** (during the prefill phase). The visual features are then stored in the KV-Cache.
>   * **Cost Formula:**
>
>     $$\text{Total Latency} \approx \underbrace{\text{Visual Interaction}}_{\text{Once (Heavy)}} + \sum\_{t=1}^{N} \text{Decoding Steps}$$
>   * **Result:** For a sequence of $N$ tokens, the model performs **1** visual interaction and **$N$** lightweight decoding steps.
>
> -----
> ### **🅱️. Diffusion Fine-Tuning (DFT): Coupled Visual Interaction ($T \times \text{Full}$)**
> In contrast, Diffusion models inherently couple visual interaction with the denoising process.
>
>   * **Mechanism:** At every denoising step, the entire noisy text sequence is updated. Therefore, the model must re-calculate the **Full Cross-Attention** between the text and visual features at every step.
>   * **Cost Formula:**
>     $$\text{Total Latency} \approx \sum_{t=1}^{T} (\underbrace{\text{Visual Interaction} + \text{Denoising}}_{\text{Heavy (Every Step)}})$$
>   * **Result:** The model performs **$T$** visual interactions. Since the per-step cost is high, we must reduce the total steps ($T$) to outperform AR.
>
> -----
>
> ## 🏗️ 2. Sensitivity Analysis: Why Multi-Token Decoding is Necessary
>
> Since DFT pays the `Visual Interaction Tax` at every step, we must reduce the total steps ($T$) to be competitive. We vary the **Effective Throughput** (tokens generated per visual interaction) to find the breakeven point.
>
> > **Table 🔴: Efficiency vs. Visual Interaction Frequency**
> > *Task: 16-Point Polygon ($N \approx 100$ tokens).*
>
>
> | Method | Visual Interactions ($T$) | Tokens/Step ($N/T$) | Latency (ms) $\downarrow$ | RefCOCO val |
> | :--- | :---: | :---: | :---: | :---: |
> | Qwen-2.5-VL (7B) | 1 | 1 | 700 | 15.5 |
> | | | | | |
> | DFT (8B) | 100 | 1 | 6000 | 21.8 |
> | DFT (8B) | 20 | 5 | 1400 | 23.0 |
> | DFT (8B) | 10 | 10 | 900 | 20.1 |
> | DFT (8B) | 6 | ~15 | 700 | 13.4 |
> | DFT (8B) | 5 | 20 | 640 | 12.4 |
>
> -----
>
>
> ### **📊 Results Discussion**
>
> * **Peak Performance vs. Over-Refinement:** The AR baseline bottlenecks at **15.5**. DFT reaches peak accuracy at **$T=20$ (23.0)**. Notably, extending to $T=100$ slightly degrades performance (21.8), as excessive denoising steps lead to **geometric collapse** (e.g., shrinking or vanishing polygons).
> * **The Optimal Trade-off ($T=10$):** This is the efficiency sweet spot. We trade a **marginal 200ms latency increase** (900ms vs 700ms) for a **substantial accuracy boost (+4.6)** compared to the baseline.
> * **Efficiency Limit:** Strictly matching AR's speed ($T=6$, 700ms) compromises accuracy (13.4), confirming that $\sim10$ iterative visual interactions are essential for precise grounding.
>
> >  **Conclusion:**
> While AR provides fixed low latency, it is limited in precision. At the optimal setting ($T=10$), DFT delivers **superior accuracy (+4.6)** at a **competitive speed**, establishing it as the better choice for high-precision tasks.

---

> ### Author Response · Authors · 2025-11-25
> **Reply to Reviewer tJXR -- 2**
>
> # `Weakness 2 & Question 2: More vertices (>16)`
>
> We thank the reviewer for noting the drop at 24 points. This stemmed from our inference scheduling strategy rather than a model limitation.
>
> ### 💉 ***1. Diagnosis: Limitations of Single-Token Decoding***
> In our initial results (Table 4), the conservative ~1 token/step strategy required excessive steps for 24-point polygons ($144$ tokens). This extended horizon caused 'drift,' leading to 'geometric collapse' (e.g., 4.5 Acc) where the shape degraded due to accumulated deviations.
>
> -----
>
> ### 🌈 ***2. Solution: Dynamic Parallel Decoding***
> We implemented Dynamic Parallel Decoding (~5 tokens/step) to enforce global consistency via joint optimization. This benefits all complexity levels, improving 16-point performance and completely resolving the collapse at 24/32 points.
>
> -----
>
> ### 🌟 ***3. New Experimental Results (16 to 32 Points):***
> We re-evaluated the model on 16, 24, and 32 vertices. The results demonstrate that DFT is highly scalable when parallel decoding is utilized.
>
> > **Table 🔴: Robustness analysis across varying polygon complexities (vertex counts).**
> >
> | Vertex Count | Model | Setting | Val | TestA | TestB |
> | :--- | :--- | :--- | :--- | :--- | :--- |
> | **16 Points** | Qwen-2.5-VL (7B) | AR Baseline | 15.5 | 16.4 | 14.9 |
> | | DFT (8B) | ~1 Token/Step | 21.8 | 22.5 | 20.9 |
> | | | **~5 Token/Step** | **23.0** | **23.8** | **22.6** |
> | **24 Points** | Qwen-2.5-VL (7B) | AR Baseline | 12.2 | 10.1 | 11.3 |
> | | DFT (8B) | ~1 Token/Step | 4.5 | 4.6 | 3.4 |
> | | | **~5 Token/Step** | **19.8** | **20.5** | **18.8** |
> | **32 Points** | Qwen-2.5-VL (7B) | AR Baseline | 11.5 | 10.2 | 11.1 |
> | | DFT (8B) | ~1 Token/Step | 2.9 | 3.7 | 3.7 |
> | | | **~5 Token/Step** | **16.4** | **17.1** | **16.6** |
>
> > **Conclusion**:
> > * **Resolved "Collapse" & Boosted Performance:** These results confirm the drop was a scheduling artifact. Parallel Decoding (~5 tokens/step) fully restores performance at 24 points (4.5 $\to$ 19.8) and boosts the 16-point baseline (21.8 $\to$ 23.0).
> > * **Superior Scalability vs. AR:** DFT consistently dominates Qwen-2.5-VL. While AR degrades with complexity (15.5 $\to$ 11.5), DFT remains robust at 32 points (16.4), effectively preventing autoregressive sequence drift.
>
>
> We further explored `tokens-per-step` settings in the inference speed section. In summary, an appropriate value should be selected based on the different lengths.

---

> ### Author Response · Authors · 2025-11-25
> **Reply to Reviewer tJXR -- 3**
>
> # `Weakness 3 & Question 2: Analysis of Limit Cases: Semantic Ambiguity, and Occlusion (specialist / AR VLM / DFT)`
>
>
> We compared DFT with specific (SeqTR) and AR LVLM (VisionLLM) baselines on limit cases, reporting both geometric (quantitative) and semantic (qualitative) results.
>
> ### 💉 ***1. Semantic Complexity & Ambiguity (Figure 15)***
>
> * **Challenge:** The model must distinguish targets based on complex logic (e.g., counting, social relations) or subtle attributes (e.g., "shiniest," "furthest"), rather than simple nouns.
> * **Visual Evidence:** As shown in the newly added **Figure 15**, baselines struggle significantly in these regimes:
>     * **Specialist Failure (SeqTR):** Without an LLM, SeqTR cannot handle complex logic. It completely misses prompts like 'The plate in front of the only member of the opposite sex facing us' by failing to parse social and spatial relations.
>
>     * **AR Generalist Failure (VisionLLM):** VisionLLM has reasoning skills but often hallucinates. In 'The vehicle of the person most likely to get rained on', it fails to link environmental context, whereas DFT correctly identifies the segway.
>
>     * **DFT Success:** DFT resolves these by fusing the LLM's reasoning with Diffusion's localization. It correctly grounds complex targets like 'The shiniest coin' or 'The second furthest horse' where others fail.
>
> ---
>
> ### 🌈 ***2. Heavy Occlusion (Figure 15 & Global Planning)***
>
> * **Challenge:** Grounding objects that are partially hidden or blocked.
> * **Visual Evidence:** In Figure 15 (Right, row 6), for the prompt *"The blocked red car in the middle"*, baselines either miss the object or focus on the occluder (the pole).
> * **DFT Success:** DFT's Global Planning overcomes AR's sequential limitations. By inferring the object's holistic structure rather than focusing on local pixels, it accurately segments the occluded car.
>
>
> > **Conclusion:**
> DFT demonstrates exceptional robustness across all three limit cases: it surpasses **Specialists** in semantic reasoning, outperforms **AR VLMs** in handling ambiguity and occlusion, and exhibits superior scalability for geometrically complex shapes.
>
> ***

---

### Official Review · Reviewer_afmP · 2025-11-01

**Soundness:** 3
**Presentation:** 2
**Contribution:** 2
**Rating:** 4
**Confidence:** 4

**Summary:**

The paper introduces Diffusion Fine-Tuning (DFT), a new framework that adapts large vision-language models for high-precision visual grounding tasks. Unlike autoregressive (AR) LVLMs (e.g., LLaVA, Qwen-VL) that predict coordinates sequentially and accumulate irreversible errors, DFT treats grounding as a global, parallel optimization. Besides, a Hierarchical Curriculum Learning strategy is adopted to progressively refine the loss supervision. Experiments show the advantages of the proposed method.

**Strengths:**

1. The paper reframes grounding from sequential prediction to diffusion-based iterative refinement. The hierarchical decomposition is intuitive and effective.

2. Training Stability. HCL convincingly stabilizes diffusion training for discrete coordinate sequences.

3. Large performance jumps (especially on polygon grounding).

4. Extension to 3D bounding box grounding (9-DoF), showing flexibility across 2D/3D domains.

5. New Benchmarks: RefCOCO-Polygon and Ref3D.

**Weaknesses:**

1. Why is the finite number of vertices (16 points) on the object’s contour chosen as the spatial localization goal instead of segmentation? Polygon grounding is less accurate and has fewer downstream applications compared with segmentation.

2. Lack of discussion and comparison with non-LVLM models. For example, PolyFormer.

3. It is unclear whether the method is based on LLaVA OneVision’s weights or LLaDA-V’s weights. The description is confusing. In the experiments, it states: “The architecture of our DFT model is adapted from a LLaVA OneVision (Zhang et al., 2024)-like autoregressive model. We therefore use it as a baseline to measure the performance gains of our diffusion paradigm.” Why is pure DLM fine-tuning not used as a comparison baseline, e.g., fine-tuning on LLaDA-V, MMaDA, or LaViDa?

4. Limited ablation scope. Most ablations isolate coordinate representation or HCL stages. There is no explicit comparison with non-hierarchical diffusion training.

**Questions:**

Please see the Weaknesses section.

One more question about HCL: if the ground truth is 299, it might be reasonable for the model to predict “3” for the hundreds digit. How does the loss function account for this?

---

> ### Author Response · Authors · 2025-11-25
> **Reply to Reviewer afmP -- 1**
>
> # `Weakness 1: Why polygon instead of instance segmentation?`
>
> We thank the reviewer. While segmentation models (e.g., LISA) offer pixel precision, we chose polygon grounding to ensure a structurally simpler, unified framework.
>
> ------
>
> ### 🦅 ***1. Structural Elegance & Unified Architecture***
>
>
> - **Complex Segmentation Pipelines**: LISA-style VLMs adopt a **pixel-to-embedding** paradigm, requiring heavy external components (alignment adapters, SAM) to decode special tokens. This complexity fractures the unified sequence generation paradigm.
>
> - **DFT's Clean, Unified Structure**: In contrast, DFT maintains a clean, decoder-free architecture (Figure 4). We treat localization purely as conditional sequence generation, directly outputting coordinates without relying on external vision modules or mask decoders.
>
> ------
>
> ### 🔎 ***2. Native Shape Perception***
>
> - Direct coordinate prediction ensures spatial reasoning remains intrinsic to the LLM, rather than being delegated to external 'drawing' decoders.
>
> - Once native perception is established, extending to segmentation is trivial (e.g., via a lightweight head). Our contribution lies in solving the core challenge of precise spatial reasoning within the generative framework.
>
>
>
> # `Weakness 2: Comparison with non-LVLM models`
>
> We thank the reviewer for suggesting the PolyFormer comparison. We will detail this in the revision; here, we highlight DFT's three critical advantages over non-LVLM specialists:
>
> ### 😷 1. Different Scopes: Generalist vs. Specialist
> * **Specialist Limitations:** Non-LVLM models (e.g., PolyFormer) are task-specific. Trained solely for grounding, they lack the broad versatility of LVLMs.
> * **DFT's Generalization:** DFT empowers General-Purpose LVLMs, preserving conversational abilities while seamlessly handling diverse localization tasks (2D, 16-point polygons, and 9-DoF 3D) within a single, unified architecture.
>
> ------
>
> ### 🧐 2. Algorithmic Superiority: Parallel vs. Autoregressive
> * **Sequential Limitation:** PolyFormer follows the standard sequential paradigm, suffering from "irreversible error accumulation" and lacking global planning capabilities.
> * **Parallel Paradigm:** DFT represents a shift to **parallel global optimization**. By refining the entire contour simultaneously, it enables inference-time error correction—a capability sequential models inherently lack.
>
> ------
>
>
> ### 📲 3. Superior Handling of Complex Semantics & Occlusion
> Unlike non-LVLM models that struggle with intricate expressions, DFT leverages LLM reasoning priors. To demonstrate these boundaries, we stress-tested DFT against **SeqTR** (Specialist) and **VisionLLM** (AR Generalist) on "limit cases," as shown in the newly added **Figure 15**.
>
>
> #### **🅰️ Semantic Complexity & Ambiguity (Figure 15)**
> * **Challenge:** Distinguishing targets based on complex logic (e.g., counting, relations) or subtle attributes, rather than simple nouns.
> * **Baseline Failures:**
>     * **SeqTR (No LLM):** Fails high-level logic. It misses relational prompts like *"The plate in front of the only member of the opposite sex facing us"* completely.
>     * **VisionLLM (Hallucination):** Fails to link context. In *"The vehicle of the person most likely to get rained on"*, it ignores the rain/environment context.
> * **DFT Success:** By fusing **LLM reasoning** with **Diffusion localization**, DFT correctly grounds complex targets like *"The shiniest coin"* or *"The second furthest horse"* where others fail.
>
> #### **🅱️ Heavy Occlusion (Figure 15 & Global Planning)**
> * **Challenge:** Grounding objects that are partially hidden or blocked.
> * **Evidence:** For the prompt *"The blocked red car in the middle"*, baselines often focus on the occluder (the pole).
> * **DFT Success:** DFT employs **Global Planning** to infer the object's holistic existence. Unlike sequential models distracted by local pixel occlusion, DFT accurately outlines the visible parts of the blocked car.
>
> ---
>
> > **Conclusion**
> > DFT demonstrates exceptional robustness: it supersedes **Specialists** (PolyFormer/SeqTR) in semantic reasoning and outperforms **AR LVLMs** in handling ambiguity and occlusion. By combining Parallel Diffusion with an LVLM foundation, DFT delivers unmatched versatility and robustness in complex scenarios.

---

> ### Author Response · Authors · 2025-11-25
> **Reply to Reviewer afmP -- 2**
>
> # `Weakness 3: Why polygon instead of instance segmentation?`
>
> We use LLaDA-V weights. Confusion may stem from LLaDA-V's architecture, which adapts LLaVA OneVision (sharing the SigLIP encoder) by replacing the AR decoder with diffusion. We cited OneVision strictly to describe this architectural foundation.
>
>
> # `Weakness 4 & Question 1: Ablation about the loss function.`
>
>
> We thank the reviewer. To validate our design and address learning efficiency (including the '299 vs 300' boundary issue), we present an ablation study across three configurations:
>
> * **Direct LLaDA-V:** The standard baseline treating coordinate prediction as a flat classification task over a 1,000-token vocabulary ($000-999$) using a single loss.
> * **Num loss LLaDA-V:** Utilizes hierarchical decomposition (Hundreds/Tens/Units) but applies simultaneous supervision to all levels, testing the representation without curriculum guidance.
> * **DFT + HCL (Ours):** Combines decomposition with **Hierarchical Curriculum Learning**. It employs a "Coarse-to-Fine" strategy, anchoring macro-outlines (Hundreds) first before progressively refining details (Tens/Units).
>
> > **Table 🔴: Ablation of Coordinate Representation & Supervision Strategy (RefCOCO-Polygon val set)**
>
> | Configuration | Supervision Strategy | Result (Acc@0.1) |
> | :--- | :--- | :---: |
> | 1. Direct LLaDA-V | Standard (Single Flat Loss) | 18.9 |
> | 2. Num loss LLaDA-V | Simultaneous Number Loss (Sum of H/T/U) | 19.3 |
> | 3. DFT + HCL (**Ours**) | Staged Curriculum (Coarse-to-Fine) | **21.8** |
>
>
> > **Conclusion**:
> > - **Decomposition alone is insufficient**: the marginal gain (+0.4) from Flat (18.9) to Simultaneous Hierarchical (19.3) confirms that structural decomposition provides limited benefit without a proper learning order.
> > - **HCL is the key driver**: the significant leap (+2.5) in Config 3 confirms that the 'coarse-to-fine' curriculum—anchoring macro-location before refining details—is decisive for precision.

---

> ### Author Response · Authors · 2025-11-28
> **Looking forward to your reply**
>
> We hope that our responses have satisfactorily addressed all your concerns and clarified the points you raised. We have made every effort to carefully consider and respond to each of your questions. We sincerely look forward to your positive feedback, as your endorsement is of great importance to us.
>
> If you find our response addresses your concerns, **we would greatly appreciate your reconsideration of the rating**.

---

### Official Review · Reviewer_EV3T · 2025-11-04

**Soundness:** 3
**Presentation:** 3
**Contribution:** 2
**Rating:** 6
**Confidence:** 3

**Summary:**

The authors propose Diffusion Fine-Tuning (DFT), reframing grounding as a robust, parallel global optimization process through a “sculpture-like” coarse-to-fine generation scheme. DFT introduces two key components: (1) Hierarchical Coordinate Decomposition (HCD), which splits coordinates into hundreds, tens, and units digits to shrink the vocabulary (1000→10) and encode spatial priors; and (2) Hierarchical Curriculum Learning (HCL), a stage-wise “easy-to-hard” training strategy with Teacher Forcing to stabilize and refine learning. DFT achieves state-of-the-art results on RefCOCO-Polygon and RefCOCO 2D grounding, and generalizes well to 9-DoF 3D grounding. Ablations highlight HCL’s importance for complex tasks, the benefit of normalized integer coordinates, and the necessity of z-axis normalization for monocular 3D detection.

**Strengths:**

* Effectively mitigates irreversible error accumulation and lack of global planning in sequential vertex prediction by enabling iterative, bidirectional refinement—akin to a “sculpture-like” coarse-to-fine generation.
* Innovative Technical Design:
  *   Hierarchical Coordinate Decomposition (HCD): Splits coordinates into hundreds, tens, and units digits, reducing vocabulary size and embedding numerical/geometric priors.
  *   Hierarchical Curriculum Learning (HCL): A three-stage training strategy that progressively supervises from macro-contour to fine details, ensuring stable and accurate convergence.

**Weaknesses:**

* The method is limited to producing a fixed number of vertices, with a notable performance drop once the count surpasses 16.
* The coordinate quantization to 0–999 may introduce discretization error, and performance is sensitive to normalization strategies.
* The diffusion-based approach to the visual grounding was first introduced by [1], which also employed a similar coarse-to-fine modeling paradigm.

[1] Cheng, Zesen, et al. "Parallel vertex diffusion for unified visual grounding." Proceedings of the AAAI Conference on Artificial Intelligence. Vol. 38. No. 2. 2024.

**Questions:**

* Advanced LLMs can already stably handle contexts up to 32k or 128k tokens. Why can't the proposed method generate more vertices?
* Will the diffusion process be more time-consuming than the autoregressive one? Could you provide a comparison of inference speed between the two methods?

---

> ### Author Response · Authors · 2025-11-25
> **Reply to Reviewer EV3T -- 1**
>
> We thank the reviewer for noting the drop at 24 points. This stemmed from our inference scheduling strategy rather than a model limitation.
>
> ### 💉 ***1. Diagnosis: Limitations of Single-Token Decoding***
> In our initial results (Table 4), the conservative ~1 token/step strategy required excessive steps for 24-point polygons ($144$ tokens). This extended horizon caused 'drift,' leading to 'geometric collapse' (e.g., 4.5 Acc) where the shape degraded due to accumulated deviations.
>
> -----
>
> ### 🌈 ***2. Solution: Dynamic Parallel Decoding***
> We implemented Dynamic Parallel Decoding (~5 tokens/step) to enforce global consistency via joint optimization. This benefits all complexity levels, improving 16-point performance and completely resolving the collapse at 24/32 points.
>
> -----
>
> ### 🌟 ***3. New Experimental Results (16 to 32 Points):***
> We re-evaluated the model on 16, 24, and 32 vertices. The results demonstrate that DFT is highly scalable when parallel decoding is utilized.
>
> > **Table 🔴: Robustness analysis across varying polygon complexities (vertex counts).**
> >
> | Vertex Count | Model | Setting | Val | TestA | TestB |
> | :--- | :--- | :--- | :--- | :--- | :--- |
> | **16 Points** | Qwen-2.5-VL (7B) | AR Baseline | 15.5 | 16.4 | 14.9 |
> | | DFT (8B) | ~1 Token/Step | 21.8 | 22.5 | 20.9 |
> | | | **~5 Token/Step** | **23.0** | **23.8** | **22.6** |
> | **24 Points** | Qwen-2.5-VL (7B) | AR Baseline | 12.2 | 10.1 | 11.3 |
> | | DFT (8B) | ~1 Token/Step | 4.5 | 4.6 | 3.4 |
> | | | **~5 Token/Step** | **19.8** | **20.5** | **18.8** |
> | **32 Points** | Qwen-2.5-VL (7B) | AR Baseline | 11.5 | 10.2 | 11.1 |
> | | DFT (8B) | ~1 Token/Step | 2.9 | 3.7 | 3.7 |
> | | | **~5 Token/Step** | **16.4** | **17.1** | **16.6** |
>
> > **Conclusion**:
> > * **Resolved "Collapse" & Boosted Performance:** These results confirm the drop was a scheduling artifact. Parallel Decoding (~5 tokens/step) fully restores performance at 24 points (4.5 $\to$ 19.8) and boosts the 16-point baseline (21.8 $\to$ 23.0).
> > * **Superior Scalability vs. AR:** DFT consistently dominates Qwen-2.5-VL. While AR degrades with complexity (15.5 $\to$ 11.5), DFT remains robust at 32 points (16.4), effectively preventing autoregressive sequence drift.
>
>
> We further explored `tokens-per-step` settings in the inference speed section. In summary, an appropriate value should be selected based on the different lengths.
>
>
> ````
>
> ````
>
> # `Weakness 2: coordinate quantization error (000–999)`
>
> ### 🧪  ***1. Resolution Sufficiency:***
> Given that standard input resolutions for LVLMs are typically around $384 \times 384$ to $1024 \times 1024$ (our implementation uses dynamic resolution with patch size 384 3), a 1000-bin quantization offers near-pixel-level precision, making the discretization error negligible compared to the gain in semantic alignment.
>
>
> ------
>
> ### 📽️ ***2. New Experiment (0-999 vs. 0-9999):***
> To empirically verify if the 0-999 quantization creates a precision bottleneck, we trained a variant of DFT with a 10x larger vocabulary (0-9999).
>
> > **Table 🔴: Ablation study on coordinate quantization granularity.**
> >
> | Coordinate Quantization | Val (Acc) | TestA (Acc) | TestB (Acc) |
> | :--- | :--- | :--- | :--- |
> | **Normed 000-999 (Ours)** | **21.8** | **22.5** | **20.9** |
> | **Normed 0000-9999** | 22.0 | 22.4 | 21.5 |
>
> > **Conclusion**:
> > * As shown in Table R2, increasing the granularity to 10,000 bins yields negligible improvement (fluctuations within $\pm 0.2\%$).
> > * The current 1,000-bin resolution (approx. 1 bin per 0.3-1 pixel depending on image size) is sufficient for the task. The primary challenge in polygon grounding remains high-level spatial reasoning rather than low-level coordinate precision.

---

> ### Author Response · Authors · 2025-11-25
> **Reply to Reviewer EV3T -- 2**
>
> # `Weakness 3: The comparision with <Parallel Vertex Diffusion>`
> Regarding PVD[1], we acknowledge its contribution but clarify fundamental differences: our approach uses Discrete (vs. Continuous) diffusion, an LLM (vs. Standard Decoder) backbone, and a distinct 'coarse-to-fine' definition.
>
>
> ### 😶‍🌫️ ***1. Discrete vs. Continuous Diffusion Paradigms***
>
> - PVD (Continuous): Models coordinates as vectors and regresses floating-point values using standard Gaussian noise.
>
> - DFT (Discrete): Operates within an LLM's discrete vocabulary. We model categorical transitions ([MASK] $\to$ digits) rather than continuous regression. This specifically addresses the "Large Vocabulary" challenge of language modeling, which PVD does not face.
>
> ------
>
> ### 🌏️ ***2. Distinct Mechanisms of Coarse-to-Fine***
>
> - PVD achieves spatial refinement via Gaussian denoising and anchor offsets in continuous space.
> - DFT employs semantic refinement via Hierarchical Coordinate Decomposition (HCD), sculpting coordinates digit-by-digit (e.g., Hundreds$\to$Tens$\to$Units). `This numerical strategy is designed specifically for LLM tokenization` and has no equivalent in PVD.
>
> ------
>
> ### ⚙️ ***3. Generalization & Semantic Depth of Model Paradigm***
>
> Finally, PVD is a ***specialized task-specific model***, whereas DFT is a ***general-purpose Multimodal Diffusion Model (MDM)*** framework. This paradigm shift offers three key advantages:
>
>
> - **Deep Semantic Understanding**: As an LLM-based model, DFT possesses deep semantic reasoning capabilities, allowing it to handle complex instructions and scenarios involving severe occlusion.
>
>
> > We added an `Appendix figure comparing DFT with SeqTR` (similar to PVD’s baseline). It demonstrates DFT's superiority on complex instructions and occlusions, where non-LLM approaches fail.
>
> - **Clean Architecture & Multi-Tasking**: DFT employs an **LLM Direct Output architecture**, avoiding PVD's specialized denoisers. This streamlines the model and preserves its versatility, enabling both high-precision grounding and multimodal tasks like VQA.
>
> - **Unified 3D Grounding Capability**: DFT unifies 2D and 3D tasks. We demonstrate that DFT achieves strong performance on `9-DoF 3D Bounding Box Grounding` (including depth prediction) across indoor and outdoor datasets. This is a capability that PVD , SeqTR, and Polyformer cannot achieve.
>
> ````
>
> ````
>
> We compared DFT with specific (SeqTR) and AR LVLM (VisionLLM) baselines on limit cases, reporting both geometric (quantitative) and semantic (qualitative) results.
>
> ### 💉 ***1. Semantic Complexity & Ambiguity (Figure 15)***
>
> * **Challenge:** The model must distinguish targets based on complex logic (e.g., counting, social relations) or subtle attributes (e.g., "shiniest," "furthest"), rather than simple nouns.
> * **Visual Evidence:** As shown in the newly added **Figure 15**, baselines struggle significantly in these regimes:
>     * **Specialist Failure (SeqTR):** Without an LLM, SeqTR cannot handle complex logic. It completely misses prompts like 'The plate in front of the only member of the opposite sex facing us' by failing to parse social and spatial relations.
>
>     * **AR Generalist Failure (VisionLLM):** VisionLLM has reasoning skills but often hallucinates. In 'The vehicle of the person most likely to get rained on', it fails to link environmental context, whereas DFT correctly identifies the segway.
>
>     * **DFT Success:** DFT resolves these by fusing the LLM's reasoning with Diffusion's localization. It correctly grounds complex targets like 'The shiniest coin' or 'The second furthest horse' where others fail.
>
> ---
>
> ### 🌈 ***2. Heavy Occlusion (Figure 15 & Global Planning)***
>
> * **Challenge:** Grounding objects that are partially hidden or blocked.
> * **Visual Evidence:** In Figure 15 (Right, row 6), for the prompt *"The blocked red car in the middle"*, baselines either miss the object or focus on the occluder (the pole).
> * **DFT Success:** DFT's Global Planning overcomes AR's sequential limitations. By inferring the object's holistic structure rather than focusing on local pixels, it accurately segments the occluded car.
>
>
> > **Conclusion:**
> DFT demonstrates exceptional robustness across all three limit cases: it surpasses **Specialists** in semantic reasoning, outperforms **AR VLMs** in handling ambiguity and occlusion, and exhibits superior scalability for geometrically complex shapes.
>
> ***

---

> ### Author Response · Authors · 2025-11-25
> **Reply to Reviewer EV3T -- 3**
>
> # `Question 2: Time-consuming compared with the autoregressive models`
>
>
> ## 📦️ ***1. Theoretical Complexity Analysis***
>
>
> We analyze the fundamental efficiency difference by examining how frequently each model performs **Visual Interaction** (Cross-Attention between text and image).
>
> -----
>
> ###  **🅰️. AR Baseline: Decoupled Visual Encoding ($1 + N$)**
> Autoregressive models decouple visual processing from text generation.
>
>   * **Mechanism:** The heavy visual interaction occurs **only once** (during the prefill phase). The visual features are then stored in the KV-Cache.
>   * **Cost Formula:**
>
>     $$\text{Total Latency} \approx \underbrace{\text{Visual Interaction}}_{\text{Once (Heavy)}} + \sum\_{t=1}^{N} \text{Decoding Steps}$$
>   * **Result:** For a sequence of $N$ tokens, the model performs **1** visual interaction and **$N$** lightweight decoding steps.
>
> -----
> ### **🅱️. Diffusion Fine-Tuning (DFT): Coupled Visual Interaction ($T \times \text{Full}$)**
> In contrast, Diffusion models inherently couple visual interaction with the denoising process.
>
>   * **Mechanism:** At every denoising step, the entire noisy text sequence is updated. Therefore, the model must re-calculate the **Full Cross-Attention** between the text and visual features at every step.
>   * **Cost Formula:**
>     $$\text{Total Latency} \approx \sum_{t=1}^{T} (\underbrace{\text{Visual Interaction} + \text{Denoising}}_{\text{Heavy (Every Step)}})$$
>   * **Result:** The model performs **$T$** visual interactions. Since the per-step cost is high, we must reduce the total steps ($T$) to outperform AR.
>
> -----
>
> ## 🏗️ 2. Sensitivity Analysis: Why Multi-Token Decoding is Necessary
>
> Since DFT pays the `Visual Interaction Tax` at every step, we must reduce the total steps ($T$) to be competitive. We vary the **Effective Throughput** (tokens generated per visual interaction) to find the breakeven point.
>
> > **Table 🔴: Efficiency vs. Visual Interaction Frequency**
> > *Task: 16-Point Polygon ($N \approx 100$ tokens).*
>
>
> | Method | Visual Interactions ($T$) | Tokens/Step ($N/T$) | Latency (ms) $\downarrow$ | RefCOCO val |
> | :--- | :---: | :---: | :---: | :---: |
> | Qwen-2.5-VL (7B) | 1 | 1 | 700 | 15.5 |
> | | | | | |
> | DFT-8B | 100 | 1 | 6000 | 21.8 |
> | DFT-8B | 20 | 5 | 1400 | 23.0 |
> | DFT-8B | 10 | 10 | 900 | 20.1 |
> | DFT-8B | 6 | ~15 | 700 | 13.4 |
> | DFT-8B | 5 | 20 | 640 | 12.4 |
>
> -----
>
>
> ### **📊 Results Discussion**
>
> * **Peak Performance vs. Over-Refinement:** The AR baseline bottlenecks at **15.5**. DFT reaches peak accuracy at **$T=20$ (23.0)**. Notably, extending to $T=100$ slightly degrades performance (21.8), as excessive denoising steps lead to **geometric collapse** (e.g., shrinking or vanishing polygons).
> * **The Optimal Trade-off ($T=10$):** This is the efficiency sweet spot. We trade a **marginal 200ms latency increase** (900ms vs 700ms) for a **substantial accuracy boost (+4.6)** compared to the baseline.
> * **Efficiency Limit:** Strictly matching AR's speed ($T=6$, 700ms) compromises accuracy (13.4), confirming that $\sim10$ iterative visual interactions are essential for precise grounding.
>
> > **Conclusion:**
> While AR provides fixed low latency, it is limited in precision. At the optimal setting ($T=10$), DFT delivers **superior accuracy (+4.6)** at a **competitive speed**, establishing it as the better choice for high-precision tasks.

---

### Note · Program_Chairs · 2026-01-17
**Submission Desk Rejected by Program Chairs**

The following references in this submission do not refer to real documents and/or have major errors in bibliographic information:

 Yuanhan Zhang, Haotian Liu, Chunyuan Li, Huilong Chen, Peng-Shuai Wang, Xinlong Wang, and Yong Jae Lee. One-Vision-Sufficient: A recipe for general-purpose vision-language models. arXiv preprint arXiv:2408.06263, 2024.